# INSPIRE: Incorporating Diverse Feature Preferences in Recourse

**Prateek Yadav**                                   *praty@cs.unc.edu*
**Peter Hase**                                       *peter@cs.unc.edu*
**Mohit Bansal**                                   *mbansal@cs.unc.edu*
*UNC-Chapel Hill*

**Reviewed on OpenReview:** *https://openreview.net/forum?id=6yzIuqKGnq*

## Abstract

Most recourse generation approaches optimize for indirect distance-based metrics like diversity, proximity, and sparsity, or a shared cost function across all users. A shared cost function in particular is an unrealistic assumption because users can have diverse feature preferences (FPs), i.e. the features they are willing to act upon to obtain recourse. In this work, we propose a novel method, INSPIRE to incorporate diverse feature preferences in both recourse generation and evaluation procedures by focusing on the cost incurred by a user when opting for a recourse. To achieve this, we first propose an objective function, *Expected Minimum Cost* (EMC) based on two key ideas: (1) the user should be comfortable adopting at least one solution when presented with multiple options, and (2) we can provide users with multiple options that cover a wide variety of FPs when the user's FPs are unknown. To optimize for EMC, we propose a novel discrete optimization algorithm, *Cost-Optimized Local Search* (COLS), that is guaranteed to improve the quality of the recourse set over iterations. Next, we propose a cost-based evaluation procedure that computes user satisfaction by simulating each user's cost function and then computing the incurred cost for the provided recourse set. Experiments on popular real-world datasets demonstrates that our method is more fair compared to baselines and satisfies up to 25.9% more users. We also show that our method is robust to misspecifications of the cost function distribution.[1]

## 1 Introduction

Over the past few years, ML models have been increasingly deployed to make critical decisions related to loan approval (Siddiqi, 2012), allocation of public resources (Chouldechova et al., 2018), and hiring decisions (Ajunwa et al., 2016). These decisions have real-life consequences for the involved users. As a result, there is a growing emphasis on explaining these models' decisions (Poulin et al., 2006; Ribeiro et al., 2018) and providing *recourse* for unfavorable decisions (Voigt & dem Bussche, 2018). A *recourse* is an actionable plan that allows a user to change the decision of a deployed model to a desired alternative (Wachter et al., 2017). Recourses are often presented to users as a set of *counterfactuals* (CFs), where each CF details the changes to the *user's state vector* (i.e., their feature vector). Recourses are desired to be *actionable*, and *feasible*. *Actionable* means that only features that can be changed by the user are requested to be changed. A recourse is *feasible* if it is actionable and easy to adopt for the user.

To achieve these objectives, prior work used indirect feature distance-based objectives like *proximity*, *sparsity*, and *feature diversity*. For instance, Mothilal et al. (2020) and Wachter et al. (2017) encourage *proximity* by minimizing the distance between the user's state vector and the counterfactuals (CFs) with the assumption that proximal CFs are easier to adopt. Whereas, *sparsity* quantifies the number of features that require modification to implement a recourse (Mothilal et al., 2020). In contrast to these, *feature diversity* (Mothilal et al., 2020; Cheng et al., 2021) provides a user with multiple CFs that change diverse subsets of features assuming

---

[1]Our code is available at https://github.com/prateeky2806/EMC-COLS-recourse.

that users are more likely to find at least one feasible solution. Any of these objectives alone is not sufficient to provide feasible recourse to the user and it is hard and subjective to decide what combination of them will lead to a satisfied user. Moreover, these objectives do not account for individual user preferences which should be of primary focus. For instance, if a user prefers to change features $f_1$ and $f_2$, then providing them with recourses that change undesirable features makes them infeasible even if they are proximal, sparse, and diverse.

To address this, some recourse methods define and use cost functions. A *cost function*, $\mathcal{C}(f, i, j)$ denotes the cost of changing a feature $f$ from value $i$ to $j$. The methods of Ustun et al. (2019); Rawal & Lakkaraju (2020); Karimi et al. (2020c;d) *assume that all the users share a single cost function.* They define this cost function and then for each user, they optimize the same cost function to generate CFs with low overall costs under this function. Next, they evaluate the cost of CFs under the same cost function. We question the utilization of a single cost function for all users, as it may not accurately reflect the preferences of a diverse user population.

We argue that it is crucial to assume that each user has a different cost function in order to effectively cater to the unique characteristics of individual users. However, as noted by Rawal & Lakkaraju (2020), in most practical cases it is difficult for a user to specify their feature preferences or cost function. Therefore, we assume that these are not provided and we only assume access to the user's state vector. To overcome this limitation, we propose a novel method, INSPIRE (INcorporating diverSe feature Preferences In RecoursE) that generates a diverse set of cost functions to capture various feature preferences (FPs) a user might possess and utilizes them to provide a recourse set containing multiple CFs to the user. INSPIRE provides each user with a recourse set that is constructed such that the likelihood of having at least one feasible solution adhering to the user's personal feature preference is maximized. INSPIRE focuses on and improves upon four major components – (1) the procedure to generate diverse cost functions with underlying FPs, (2) the recourse objective function, (3) the recourse generation algorithm, and (4) the evaluation procedure.

We build on the cost functions defined in Ustun et al. (2019) to propose three distributions, $\mathcal{D}_{step}$, $\mathcal{D}_{perc}$, and $\mathcal{D}_{mix}$ that can be used to

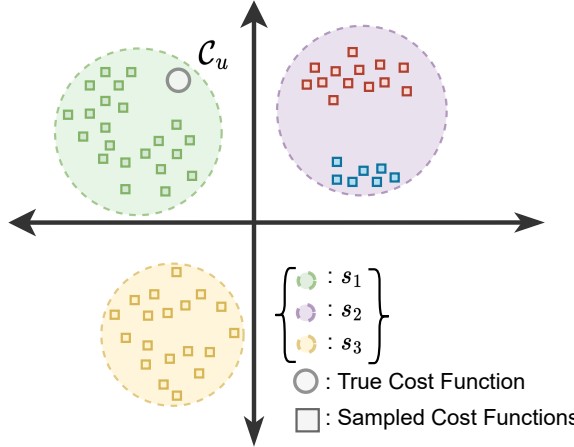

Figure 1: Diagram showing the intuition behind the Expected Minimum Cost Objective. The axes represent an abstract cost function space where squares denote cost function samples that are the same color if they have similar underlying FPs and hence form a cluster. We aim to find a recourse set where each CF (here, $\{s_1, s_2, s_3\}$) does well for certain types of FPs represented by clusters of cost functions. The shaded big circles each represent a single CF $s_i$ that caters to the enclosed cost functions. Here the user's ground-truth cost function (grey circle) is served well by $s_1$.

sample cost functions with different underlying FPs that users in a population might possess. These distributions generate cost functions based on steps needed and percentile changes (Ustun et al., 2019) in the feature values (§4.1). Next, we propose a novel objective function, *Expected Minimum Cost* (EMC) that approximately optimizes the cost a user will incur when their FPs are unknown. The EMC objective encourages diversity in the solution set with respect to the different FPs a user might possess. This is done by ensuring that each CF is a good CF under some particular cluster of cost functions representing similar FPs (see Figure 1). We minimize the EMC of the generated recourse set with respect to multiple sampled cost functions from one of the proposed distributions (§4.2). Hence, if the user's ground-truth cost function is well represented by any of the clusters, then we will have *some* counterfactual that is feasible (actionable and low-cost) for them (See Figure 1). In order to efficiently optimize for EMC, we propose a genetic algorithm (Mitchell, 1998) based optimization method, *Cost-Optimized Local Search* (COLS) (§4.3). COLS guarantees a monotonic reduction in EMC of the recourse set, leading to large empirical reductions in the user-incurred cost.

Lastly, we propose a novel cost-based evaluation procedure to compare different methods that accounts for the individuality of the users by having different cost functions for each user (§4.4). We simulate an evaluation cost function for each user and then assess the *fraction of satisfied users* based on whether their incurred cost of recourse is below a certain satisfiability threshold $k$. Even though the cost functions are simulated, this setup can be used to rank different methods to assess their recourse generation abilities. We also report *coverage*, which is the fraction of users with at least one actionable recourse (Rawal & Lakkaraju, 2020). Moreover, we evaluate all methods on existing indirect metrics from the literature like diversity, proximity, sparsity, and validity (§5.1).

To evaluate the effectiveness of EMC and COLS, we run experiments on two popular real-world datasets: Adult-Income (Dua & Graff, 2017) and COMPAS (Larson et al., 2016). We compare our method with multiple popular and recent baseline methods like DICE (Mothilal et al., 2020), FACE (Poyiadzi et al., 2020), and Actionable Recourse (AR) (Ustun et al., 2019). We show that our method satisfies up to 25.89% more users than strong baseline methods while covering up to 22.35% more users across datasets. We also perform important ablations to show the advantages of using the EMC objective and the COLS optimization procedure. Next, we perform a fairness analysis of methods across demographic subgroups to show that our method is fairer than baseline methods. Finally, we evaluate the robustness of our method to various types of distribution shifts that can occur between the evaluation cost functions distribution and the distributions used in the EMC objective (see Section 5). We find that our method is robust to multiple types of distribution shifts and leads to an improvement of at least 11% even in such scenarios. Our primary contributions are listed below.

1. We conceptualize a novel method, INSPIRE that accounts for diverse feature preferences while generating and evaluating recourse options. INSPIRE provides the flexibility for future researchers to further innovate on all of its four components.
2. We propose a new objective function, Expected Minimum Cost that approximately optimizes the cost a user will incur when their FPs are unknown using diverse samples from a distribution.
3. We propose a discrete optimization method, *Cost-Optimized Local Search* which generates recourses that lead to higher user satisfaction while being fairer.
4. We propose an evaluation procedure that allows us to compare different methods on individual user satisfaction by simulating users' evaluation cost functions.

## 2 Related Work

Here, we distinguish our approach based on our recourse objectives, optimizer, and evaluation. We point readers to Venkatasubramanian & Alfano (2020) for a philosophical basis of algorithmic recourse and to Karimi et al. (2020b) for a comprehensive survey of the existing recourse methods.

**Objectives:** The most prominent family of objectives for recourse includes distance-based objectives (Wachter et al., 2017; Karimi et al., 2020a; Dhurandhar et al., 2018; Mothilal et al., 2020; Rasouli & Yu, 2021). These methods primarily seek recourses that are close to the original data point. In DICE, Mothilal et al. (2020) provides users with a set of counterfactuals while trading off between proximity and feature diversity. A second category of methods uses other heuristics based on the data distribution (Aguilar-Palacios et al., 2020; Gomez et al., 2020) to come up with counterfactuals. FACE constructs a graph from the given data and then tries to find a high-density path between points in order to generate counterfactuals (Poyiadzi et al., 2020). Lastly, the works closest to ours are the cost-based objectives, which capture feasibility in terms of the cost of recourse: (1) Cui et al. (2015) define a cost function specifically for tree-based classifiers, which compare the different paths that two data points follow in a tree to obtain a classifier-dependent measure of cost. (2) Karimi et al. (2020c;d) take a causal intervention perspective on the task and define cost in terms of the normalized distance between the user state and the counterfactual. (3) Ustun et al. (2019) define cost in terms of the number of changed features and frame recourse generation as an Integer Linear Program. (4) Rawal & Lakkaraju (2020) infer global cost function from pairwise comparisons of features that are drawn from simulated users. However, they take a different approach to the recourse generation problem, which is to find a list of rules that can apply to any user to obtain a recourse, rather than especially generating recourses for each user as in this work. Importantly, all of these works assume there is a known and single cost function that is shared by all users.

**Optimization:** Several recourse methods use gradient-based optimization to generate counterfactuals close to a user's data point (Wachter et al., 2017; Mothilal et al., 2020). Some recent approaches use tree-based techniques (Rawal & Lakkaraju, 2020; von Kügelgen et al., 2020; Kanamori et al., 2020) or kernel-based methods (Dandl et al., 2020; Gomez et al., 2020; Ramon et al., 2020), while others employ some heuristic (Poyiadzi et al., 2020; Aguilar-Palacios et al., 2020) to generate counterfactuals. A few works use autoencoders to generate recourses (Pawelczyk et al., 2020; Joshi et al., 2019), while (Karimi et al., 2020a) and (Ustun et al., 2019) utilize SAT and ILP solvers, respectively. In contrast to these methods, we also suspect that ideas from distributionally robust optimization (Rahimian & Mehrotra, 2019) can also be used to come up with alternative objective functions or even better ways to optimize the EMC objective.

**Evaluation:** Besides ensuring recourse validity, the most prominent approaches to evaluate recourses rely on distance-based metrics. In DICE, Mothilal et al. (2020) evaluate recourses according to their proximity, sparsity, and feature diversity. Meanwhile, several works directly consider the cost of the recourses, using a single known cost function as a metric, meaning that all users share the same cost function. In contrast, Rawal & Lakkaraju (2020) estimate a single rank order amongst the feature from simulated pairwise feature comparisons from experts. For all these methods, a single cost function is used for both recourse generation and evaluation, i.e. the solutions are optimized and tested on the same cost function (Cui et al., 2015; Karimi et al., 2020c;d; Rawal & Lakkaraju, 2020). In contrast, we evaluate recourse methods by simulating user-specific cost functions that can vary greatly across users to capture their preferences.

## 3 Problem Statement

**Features Types.** We assume a dataset with features $\mathcal{F} = \{f_1, f_2, ...f_h\}$. Features can be mutable, conditionally mutable, or immutable, according to the causal processes that generate them. For example, *Race* is an immutable feature (Mothilal et al., 2020), *Age* and *Education* are conditionally mutable (cannot be decreased), and *number of work hours* is mutable (can both increase and decrease). Following (Ustun et al., 2019), continuous features are always discretized into appropriately sized bins. Moreover, the feature values can be ordered or unordered, e.g., *Workclass* can be business, private, government, etc and none of these feature values are more difficult then the other to achieve.

**Cost Function.** In this work, we assume that each user has an inherent *feature-preference* (FP) that is an ordering of the features with respect to the ease of changing them, and different users will likely have different FPs. We quantify these user FPs via *feature-scores* denoted by $\boldsymbol{p} = [p^{f_1}, \ldots, p^{f_h}]$, which sum to 1 and each $p^{f_i} \in [0, 1]$ represents the willingness of the user to change feature $f_i$. A cost function $\mathcal{C}(f, x, y)$ takes into account the user FPs and provides an elaborate cost of changing a feature $f$ from $x$ to $y$ and lies in $[0, 1] \cup \{\infty\}$. Here, 0 means that the transition has no associated cost, whereas 1 means it is maximally difficult, and $\infty$ means that it is infeasible under the cost function.

**Transition Costs.** Given a cost function $\mathcal{C}$ and two state vectors $\boldsymbol{s}_i, \boldsymbol{s}_j$, the cost of transition from $\boldsymbol{s}_i \to \boldsymbol{s}_j$ is the sum of the cost of changing individual features. Hence, $\text{Cost}(\boldsymbol{s}_i, \boldsymbol{s}_j; \mathcal{C}) = \sum_{f \in \mathcal{F}} \mathcal{C}(f, s_i^f, s_j^f)$, where $s^f$ is the value of feature $f$ in the state vector $\boldsymbol{s}$. Cost can be greater than 1 when changing multiple features.

**User Definition.** A user is defined as a tuple $\boldsymbol{u} = (\boldsymbol{s}_u, \mathcal{C}_u^*)$, where $\boldsymbol{s}_u$ is the user's current state vector of length $|\mathcal{F}|$ containing their feature values and $\mathcal{C}_u^*$ is their ground-truth cost function. See Appendix Table 8 for examples of $\boldsymbol{s}_u$ and FPs. Next, we define the *cost incurred by a user* when acting on a recourse set $\mathcal{S}$. As a rational user will select the least costly option, the cost they will incur is the *minimum* transition cost across all CF in the recourse set, defined as,

$$\text{MinCost}(\boldsymbol{s}_u, \mathcal{S}; \mathcal{C}_u^*) = \min_{s_j \in \mathcal{S}} \text{Cost}(\boldsymbol{s}_u, \boldsymbol{s}_j; \mathcal{C}_u^*). \qquad (1)$$

**Problem Definition.** For a user $\boldsymbol{u}$, our goal is to find a recourse set $\mathcal{S}_u$ such that there exists at least one low-cost CF with the desired outcome. Hence, if the user's ground-truth cost function $\mathcal{C}_u^*$ is provided then we can provide them with a good recourse by directly optimizing for,

$$\mathcal{S}_u = \underset{\mathcal{S}}{\arg\min} \ \text{MinCost}(\boldsymbol{s}_u, \mathcal{S}; \mathcal{C}_u^*) \quad \text{s.t.} \ \exists \ s_i \in \mathcal{S} \ \text{ and } \ F(s_i) = 1, \qquad (2)$$

where $F$ is the underlying ML model assigning a decision and 1 is the desired class user wants to achieve. In general, $F(s_i)$ can take multiple different values denoting different class a user can be categorized into. In our experiments, we restrict to two classes as Actionable recourse (Ustun et al., 2019) method only works with two classes. Moreover, we note that in practice it is difficult for a user to precisely quantify their FP and cost function. Therefore, in most practical scenarios $\mathcal{C}_u^*$ is not provided, and hence we use the EMC objective account for user FPs (§ 4.1,§ 4.2).

## 4  INSPIRE: Integrating Diverse Feature Preferences in Recourse

To generate recourse for a user, we utilize their state vector as the sole input. We begin by using our proposed distributions (§4.1) to sample multiple potential cost functions that may align with the FPs of the user. Subsequently, we compute our EMC objective for the candidate recourse set with respect to the sampled cost functions (§4.2). Then, our COLS optimization method generates an updated candidate recourse set (§4.3). Then this process is repeated iteratively to arrive at a final candidate recourse set, which is evaluated through our proposed evaluation procedure (§4.4).

### 4.1  Characterising Feature Preferences and Cost Function Distributions

This section aims to enhance the existing ways of defining cost functions by incorporating FPs by scaling the cost of transition with feature scores. Recent works like Ustun et al. (2019) argue that users in a population fundamentally quantify the transition cost of changing a feature $f$ from $x$ to $y$ as being proportional to – (1) the difference in percentile of $x$ and $y$, i.e. to $|Q_f(x) - Q_f(y)|$, where $Q_f(.)$ is the CDF of the feature $f$ in the target population (Ustun et al., 2019), and (2) the number of steps between to go from $x$ to $y$ (Ustun et al., 2019). For instance, when changing the education feature from *Bachelors* to *Ph.D.* the percentile might be appropriate as very few people have Ph.D. degrees as compared to Bachelors leading to a higher cost. In contrast, when changing the number of working hours from 30 to 35 users might associate a fixed cost for every additional hour as opposed to percentile differences. We call these the *Percentile* and the *Step* based transition cost. In Algorithm 2, and 3, we define procedures to sample cost function that adheres to percentile and step-based transition costs respectively. Moreover, these algorithms assign different costs based on feature properties – like being ordered or unordered, if the feature is ordered then can it both just increase, just decrease, or both increase and decrease, and if the feature is mutable, conditionally mutable, immutable. For Example, if a feature is ordered and can only increase then transition to any lower value leads to an infinite cost whereas transitioning to a higher value depends on either the percentile difference (percentile cost) or the number of steps (step cost) between the initial and final feature value. More details can be found in Algorithm 2, and 3. We note that both of these costs are monotonic, i.e., more drastic changes have higher associated costs and are useful depending on the scenario. However, these functions assign equal importance to all the features which is suboptimal.

Therefore, to incorporate the notion of FP in costs, we use the *feature scores*, $\boldsymbol{p}$ (see §3) to scale these costs. We scale the transition costs of each feature $f$ by $(1 - p^f)$, i.e Cost$(f, ., .) * (1 - p^f)$, resulting in a decreased cost for preferred features and vice-versa. This creates cost functions adhering to feature score $\boldsymbol{p}$.

Ideally, we want the user to provide us with the feature scores $\boldsymbol{p}$ but Rawal & Lakkaraju (2020) highlighted that even for experts, it is hard to specify their feature scores. Hence, to generate diverse cost functions, we want a procedure to sample feature scores $\boldsymbol{p}$ which can represent diverse FPs that users in a population might possess. Therefore, in Algorithm 4, we propose a feature score sampling procedure with minimal assumptions. We first randomly sample a subset of FPs and then for the selected features, we sample the feature scores $\boldsymbol{p}$ from a Dirichlet distribution. Note, that this procedure is very customizable, and in cases where users can specify their FPs or feature scores, we can skip this step.

Next, we combine the feature score sampling procedure (Algorithm 4) with percentile (Algorithm 2) and step (Algorithm 3) based cost function generation procedures respectively and call them $\mathcal{D}_{perc}$, $\mathcal{D}_{step}$ distributions. Finally, we propose $\mathcal{D}_{mix}$ (Algorithm 5) as our most general distribution that takes a convex combination of both step and percentile-based costs using a *cost-type* weight $\alpha$. Hence, given a state vector $\boldsymbol{s}$ and feature scores $\boldsymbol{p}$, we can use any of these three distributions to generate a cost function with an underlying FP. Note

that these distributions are using existing concepts of step and percentile-based cost functions along with feature score-based scaling. This allows us to cover a much larger space of cost functions compared to past works that assume a single fixed cost function.

## 4.2 Expected Minimum Cost Objective Function

As noted by Rawal & Lakkaraju (2020), in most practical scenarios the user's true cost-function $\mathcal{C}_u^*$ is hard to obtain thus we cannot exactly minimize for Equation 2. Hence, we propose the *Expected Minimum Cost* (EMC) objective function that utilizes samples of cost functions that depict a variety of underlying FPs that may be close to the user's true cost function, in order to generate a recourse set with at least one low-cost CF for all of these sampled cost functions (see Figure 1). Give a state vector $s$, a recourse set $\mathcal{S}$, and a distribution $\mathcal{D}_{train}$, we compute EMC as follows,

$$\text{EMC}(s, \mathcal{S}) = \mathbb{E}_{\mathcal{C}_i \sim \mathcal{D}_{train}}[\text{MinCost}(s, \mathcal{S}; \mathcal{C}_i)] \approx \frac{1}{M} \sum_{i=1}^{M} \min_{s_j \in \mathcal{S}} \text{Cost}(s, s_j; \mathcal{C}_i). \tag{3}$$

We employ Monte Carlo Estimation (Robert & Casella, 2010) to approximate the expectation by sampling $M$ cost functions $\{\mathcal{C}_i\}_{i=1}^{M}$ from $\mathcal{D}_{train}$ and then expand MinCost using Equation 1. Next, for the user $u$, we obtain the recourse set $\mathcal{S}_u$ by minimizing the EMC objective as follows,

$$\mathcal{S}_u = \arg\min_{\mathcal{S}} \; \text{EMC}(s_u, \mathcal{S}) \quad \text{s.t.} \;\; \exists \; s \in \mathcal{S} \;\; \text{and} \;\; F(s) = 1, \tag{4}$$

## 4.3 Cost Optimized Local Search (COLS)

To generate a recourse set $\mathcal{S}_u$ for a user $u$, we optimize for EMC as shown in Equation 4. We propose two simple, and efficient genetic algorithms (Mitchell, 1998) for discrete search (Pirlot, 1996), namely *Cost-Optimized Local Search* (COLS) and *Parallel Cost-Optimized Local Search* (P-COLS) presented in Algorithm 1. COLS maintains a best set $\mathcal{S}^{best}$ which will be the final recourse provided to the user. We initialize all the elements of $\mathcal{S}_0$ as $s_u$. At each iteration $t$, a candidate set $\mathcal{S}_t$ is generated by locally perturbing each element of the best set with a Hamming distance of two, i.e. making small changes to two features at once. For features with no ordering, we uniformly select any other potential value, whereas, for features with a natural ordering we select values that are closest to the original values. Then we assess the validity of the generated CFs and do not use invalid CFs to compute metrics or in any subsequent steps. However, for simplicity for the pseudocode, we omit such details. Then $\mathcal{S}_t$ is evaluated against the EMC objective as defined in Equation 3. For efficient implementation, we store the costs of all the $N$ CFs with respect to all the

---

**Algorithm 1** Cost-Optimized Local Search Algorithm

**Input:** A state vector $s$, $\{\mathcal{C}_i\}_{i=1}^{M} \sim \mathcal{D}_{train}$ cost distributions
**Output:** $\mathcal{S}^{best}$, a set with $N$ generated counterfactuals.
**function** COLS($s, \{\mathcal{C}_i\}_{i=1}^{M}$, Budget, hamDist)
  **Initialize**
    // Perturb $s$, $N$ times.
    $\mathcal{S}^{best} \in \mathbb{R}^{N \times d} \leftarrow \text{pertubCFS}(s, \text{hamDist})$
    // Incurred costs for $\mathcal{S}^{best}$.
    $\mathbf{C}^b \leftarrow \text{getCostMatrix}(s, \mathcal{S}^{best}; \{\mathcal{C}_i\}_{i=1}^{M})$
    $t = 0; \text{maxIter} = \text{Budget}//N$
  **while** $t < maxIter$ **do**
    $\mathcal{S}_t \in \mathbb{R}^{N \times d} \leftarrow \text{pertubCFS}(\mathcal{S}^{best}, \text{hamDist})$
    $\mathbf{C} \in \mathbb{R}^{N \times M} \leftarrow \text{getCostMatrix}(s, \mathcal{S}_t; \{\mathcal{C}_i\}_{i=1}^{M})$
    $t \mathrel{+}= 1$

    // $B_{ij}$ = Change in cost when $\mathcal{S}^{best}[i] \leftarrow \mathcal{S}_t[j]$. See Algorithm 6.
    $\mathbf{B} \in \mathbb{R}^{N \times N} \leftarrow \text{computeBenefits}(\mathbf{C}^b, \mathbf{C})$   ▷ See App. Code Block 1

    // If any swaps lead to cost-benefit then greedily select which pairs to swap between $\mathcal{S}^{best}$ and $\mathcal{S}_t$ given B such that the cost can only reduce. See App. Code Block 2
    $\text{replaceIndices} \leftarrow \text{getReplaceIdx}(\mathbf{B})$
    // Swap these pairs and update $C^b$.
    **forall** $originalIdx, replaceIdx \in replaceIndices$ **do**
      $\mathcal{S}^{best}[\text{originalIdx}] = \mathcal{S}[\text{replaceIdx}]$
    **end**
    $\mathbf{C}^b \leftarrow \text{getCostMatrix}(s, \mathcal{S}^{best}; \{\mathcal{C}_i\}_{i=1}^{M})$
  **end**
  **return** $\mathcal{S}^{best}, C^b$
**end**

---

$M$ cost functions $\mathcal{C}_i$. Instead of making a direct comparison of EMC for the best-set-so-far $\mathcal{S}_{t-1}^{best}$ and the candidate set $\mathcal{S}_t$, we evaluate whether any CFs from the candidate set $\mathcal{S}_t$ would improve the EMC of the best

set $\mathcal{S}_{t-1}^{best}$ if we swapped out individual CFs. Specifically, if the benefit of replacing $s_i \in \mathcal{S}_t$ with $s_j \in \mathcal{S}^{best}$ is positive, i.e., reduces EMC of $\mathcal{S}^{best}$ then we make the replacement (see Algorithm 6 to see how we estimate benefit). Algorithm 6 initializes the benefit matrix $B$ with zeros. For each sampled cost function (there are $M$ of them), the algorithm finds the indices of the best and second-best counterfactuals. Then we consider all pairs of counterfactuals ($p, q$ pairs), p from $\mathcal{S}_{t-1}^{best}$ and q from $\mathcal{S}_t$ and checks the benefit of replacing one with another. For each counterfactual, the algorithm checks all cost functions where the $p^{th}$ counterfactual is the cheapest. If the cost of the $p^{th}$ counterfactual in $C$ is greater than that in $C^b$, the benefit of replacement is the difference in costs $(C_{pr}^b - C_{qr})$. Otherwise, the benefit is calculated as the cost of the second-best counterfactual from $C^b$ minus the minimum of the current cost and the cost of the second-best counterfactual in $C^b$ for the $r^{th}$ cost function. We then greedily replace the CFs in $\mathcal{S}_{t-1}^{best}$ with cfs from $\mathcal{S}_t$ to obtain $\mathcal{S}_t^{best}$ which has either similar or better cost than $\mathcal{S}_{t-1}^{best}$. See Algorithm 1 or Code Block 2 for details. The ability to assess the benefit of each candidate CF is critical because it allows us to constantly update the best set using CFs from a candidate set instead of waiting for an entire candidate set with lower EMC. For objectives like feature diversity, evaluating the benefit is expensive (see Appendix B.1). Moreover, for COLS we can guarantee that the EMC of the best set will monotonically decrease over time, which is formalized below:

**Theorem 4.1** (Monotonicity of COLS Algorithm). *Given the best set, $\mathcal{S}_{t-1}^{best} \in \mathbb{R}^{N \times d}$, the candidate set at iteration t, $\mathcal{S}_t \in \mathbb{R}^{N \times d}$, the matrix $\boldsymbol{C}^b \in \mathbb{R}^{N \times M}$ and $\boldsymbol{C} \in \mathbb{R}^{N \times M}$ containing the incurred cost of each counterfactual in $\mathcal{S}_{t-1}^{best}$ and $\mathcal{S}_t$ with respect to all the M sampled cost functions $\{\mathcal{C}_i\}_{i=1}^M$, there always exist a $\mathcal{S}_t^{best}$ constructed from $\mathcal{S}_{t-1}^{best}$ and $\mathcal{S}_t$ such that*

$$EMC(\boldsymbol{s}_u, \mathcal{S}_t^{best}; \{\mathcal{C}_i\}_{i=1}^M) \leq EMC(\boldsymbol{s}_u, \mathcal{S}_{t-1}^{best}; \{\mathcal{C}_i\}_{i=1}^M)$$

For the proof of the theorem, please refer to Appendix B.2.2.

**P-COLS:** The P-COLS method is a variant of COLS that starts multiple parallel runs of COLS with different initial sets. With a given computational budget, each run is allocated a fraction of the total budget. The recourse set of the run with the least EMC value is provided to the user.

## 4.4 Evaluation Procedure and Cost Based Metrics

Given that users' ground-truth cost functions $\mathcal{C}_u^*$ are unknown, it is hard to computationally compare different methods for user satisfaction and metrics like diversity, proximity, and sparsity do not directly measure this. To address this issue, for every user $\boldsymbol{u}$, we propose to simulate an *evaluation* cost function $\mathcal{C}_u^\#$ from a distribution $\mathcal{D}_{test}$. We can use Equation 1 to compute the incurred cost under $\mathcal{C}_u^\#$ which can be used to rank order different recourse methods. Note that, no method is allowed to use $\mathcal{C}_u^\#$ when generating recourse and this is exclusively kept for comparing methods. The effectiveness of this evaluation procedure increase as the distribution $\mathcal{D}_{test}$ becomes more expressive in terms of accounting for user FPs as it would align better with real users.

Given a set of users $\mathcal{U}$, each user $\boldsymbol{u} \in \mathcal{U}$ is provided with a recourse set $\mathcal{S}_u$. We compute the MinCost of $\mathcal{S}_u$ given the user's state vector $\boldsymbol{s}$ and their evaluation cost function as $\text{MinCost}(\boldsymbol{s}_u, \mathcal{S}_u; \mathcal{C}_u^\#)$. Given this MinCost, we report multiple aggregated versions of it for comparing other methods.

**Population Average Cost:** We report the Population Average Cost (PAC), that is the average cost of recourse for users in the population, defined as $\text{PAC} = \frac{1}{|\mathcal{U}|} \sum_{u \in \mathcal{U}} \text{MinCost}(\boldsymbol{s}_u, \mathcal{S}_u; \mathcal{C}_u^\#)$.

**User Coverage:** Next, we follow Rawal & Lakkaraju (2020) and define *Coverage (Cov)* that counts the fraction of users in the population who were provided with at least one actionable recourse. Given a set of users $\mathcal{U}$ and the recourse sets $\{\mathcal{S}_u\}_{u \in \mathcal{U}}$ provided to them, Coverage is defined as,

$$Cov(\mathcal{U}, \{\mathcal{S}_u\}_{u \in \mathcal{U}}) = \frac{1}{|\mathcal{U}|} \sum_{u \in \mathcal{U}} \mathbb{1}\{\text{MinCost}(\boldsymbol{s}_u, \mathcal{S}_u; \mathcal{C}_u^\#) < \infty\}.$$

**Fraction of Users Satisfied (FS@$k$):** Finally, we note that PAC cannot be used to assess individual users' satisfaction. Hence, we go a step further and introduce a new cost-based metric that directly aggregates based on each user's satisfaction. We say that a user is *satisfied* if the MinCost is below a certain acceptability

Table 1: Recourse method performance across various cost and distance metrics (Section 5.1). The numbers reported are averaged across 5 different runs. "**-**" means that higher or lower values are not necessarily better when looking at individual metrics.

| Data | Method | Metrics | | | | | | |
|------|--------|---------|--|--|--|--|--|--|
| | | Cost Metrics | | | Indirect Metrics | | | |
| | | FS@1($\uparrow$) | PAC($\downarrow$) | Cov($\uparrow$) | Div(-) | Prox(-) | Spars(-) | Val($\uparrow$) |
| | DICE | 2.47 | 1.37 | 8.32 | 3.90 | 65.80 | 47.20 | 97.90 |
| | Face-Eps | 15.23 | 0.76 | 22.60 | 4.75 | 92.22 | 74.98 | **100.0** |
| | Face-Knn | 25.30 | 0.74 | 35.00 | 8.62 | 89.07 | 71.85 | **100.0** |
| Adult-Income | Act. Recourse | 49.93 | 0.55 | 56.85 | 18.38 | 74.68 | 73.57 | 78.67 |
| | COLS | 72.57 | **0.38** | 76.07 | 25.77 | 80.22 | 76.48 | 97.15 |
| | P-COLS | **75.82** | 0.40 | **79.20** | 25.57 | 81.67 | 78.00 | 94.78 |
| | DICE | 0.40 | 0.54 | 0.40 | 11.30 | 65.00 | 32.00 | 98.90 |
| | Face-Eps | 12.20 | 0.29 | 12.20 | 2.50 | 94.20 | 60.60 | **100.0** |
| | Face-Knn | 12.20 | 0.29 | 12.20 | 2.60 | 94.10 | 60.60 | **100.0** |
| COMPAS | Act. Recourse | 65.80 | 0.40 | 66.60 | 11.87 | 80.53 | 74.07 | 44.23 |
| | COLS | 82.23 | **0.24** | 82.23 | 29.32 | 77.82 | 70.05 | 95.48 |
| | P-COLS | **83.73** | **0.24** | **83.73** | 29.38 | 78.48 | 71.30 | 92.78 |

threshold $k$. Formally, we define the *fraction of users satisfied* at a satisfiability threshold $k$ as:

$$FS@k(\mathcal{U}, \{\mathcal{S}_u\}_{u \in \mathcal{U}}) = \frac{1}{|\mathcal{U}|} \sum_{u \in \mathcal{U}} \mathbb{1}\{\text{MinCost}(\boldsymbol{s}_u, \mathcal{S}_u; \mathcal{C}_u^{\#}) < k\} \tag{5}$$

Reporting FS@$k$ is similar to reporting accuracy as opposed to test loss. This is better for comparing methods because it aggregates at an example level before performing averaging. In reality, the $k$ can vary from user to user but we keep $k$ fixed across users in our experiments because the goal of any method is to find low-cost recourses regardless of $k$. Moreover, any fixed $k$ can be used to rank-order different methods. In deployment scenarios, reasonable values of $k$ can be estimated by doing a user survey.

# 5 Experiments

## 5.1 Experimental Setup

**Dataset:** We use the Adult-Income (Dua & Graff, 2017) and COMPAS (Larson et al., 2016) datasets with Open Data Commons PDDL license. The Adult-Income dataset is based on the 1994 US Census data and contains 12 features. The model has to predict whether an individual's income is over $50,000$. COMPAS contains 7 features and was collected by ProPublica and contains information about the criminal history of defendants for analyzing recidivism. The model needs to decide bail based on predicting which applicants will recidivate in the next two years. These datasets are anonymized to prevent privacy. We preprocess both datasets based on a previous analysis where categorical features are binarized (Pawelczyk et al., 2021).[2] Our black-box model is an Multi-Layer Perceptron with 2-layers. Please refer to Appendix Tables 10 and 11 for experiments with logistic regression and Appendix A.3 and Table 5 for further experimental details.

**Baselines:** We compare our methods COLS and P-COLS with DICE (Mothilal et al., 2020), FACE-Knn and FACE-Epsilon (Poyiadzi et al., 2020), and Actionable Recourse (Ustun et al., 2019). Importantly, **we control for compute across methods** by restricting the number of forward passes to the black-box model, which are needed to decide if a counterfactual produces the desired class. For most big models, this is the rate-limiting step for each method. We ran our experiments on a local server using a single Nvidia 1080 Ti GPU. We set a fixed budget of 5000 model queries, a set size $|\mathcal{S}| = 10$, and the number of cost function samples $M = 1000$ for all methods. For a description of the objective function and other details of these baselines refer to Appendix B.1.1, B.2.3.

**Distance Based Recourse Metrics:** We also report indirect metrics like feature diversity, proximity, sparsity, and validity that are used in past works. We report the average of these metrics in percentage across all users. These metrics are a proxy for user satisfaction and any individual metric might not

---

[2] The code for the Actionable Recourse method (Ustun et al., 2019) requires binary categorical variables.

Table 2: Ablation on Search algorithms and objective functions.

| Search Alg. | Objective | Cost Metrics | | | Indirect Metrics | | |
|---|---|---|---|---|---|---|---|
| | | FS@1($\uparrow$) | PAC($\downarrow$) | Cov($\uparrow$) | Div(-) | Prox(-) | Spars(-) |
| LS | Sparsity | 10.1 | 1.304 | 29.0 | 42.7 | 66.2 | 55.8 |
| LS | Proximity | 9.7 | 1.275 | 27.0 | 42.1 | 67.5 | 55.0 |
| LS | Diversity | 0.3 | 2.393 | 7.6 | **53.3** | 50.8 | 35.6 |
| LS | EMC | 49.8 | 0.597 | 59.1 | 37.8 | 73.3 | 67.5 |
| COLS | EMC | **68.8** | **0.391** | **72.6** | 27.1 | **77.5** | **73.5** |

strongly correlate with the satisfaction of a user. For example, a recourse might have really high diversity but low proximity resulting in the recourse being infeasible. For a single user, Proximity is defined as $prox(\boldsymbol{x}, \mathcal{S}) = 1 - \frac{1}{|\mathcal{S}|} \sum_{i=1}^{|\mathcal{S}|} dist(\boldsymbol{x}, \mathcal{S}_i)$, where $\mathcal{S}_i$ is a counterfactual. Sparsity (Mothilal et al., 2020) is defined as $spar(\boldsymbol{x}, \mathcal{S}) = 1 - \frac{1}{|\mathcal{S}| * d} \sum_{i=1}^{|\mathcal{S}|} \sum_{j=1}^{|\boldsymbol{x}|} \mathbb{1}_{\{x_j \neq \mathcal{S}_{ij}\}}$. Feature diversity (Mothilal et al., 2020) is defined as $div(\mathcal{S}) = \frac{1}{Z} \sum_{i=1}^{|\mathcal{S}|-1} \sum_{j=i+1}^{|\mathcal{S}|} dist(\mathcal{S}_i, \mathcal{S}_j)$, where $Z$ is the number of terms in the double summation. Validity is defined as $val(Y) = \frac{|\{\text{unique } s_i \in \mathcal{S} \ : \ f(s_i) = +1\}|}{|\mathcal{S}|}$.

**Sampling Cost Functions for Real World Practitioner:** Real-world practitioners can design a system where they can ask each user to provide either the features ($\mathcal{F}_p$) that they find easy to change (which can be converted to preference scores using Algorithm 4 or directly provide a preference score ($p$) for each feature if they can. Given this our Algorithm 5 can be used to sample cost functions that are a better estimate of the user's real cost function. Note that these cost functions are still estimates and try to capture different ways a user can think of cost, i.e. in terms of percentiles or steps (absolute change in feature). The practitioner can use these more aligned samples to optimize the EMC objective to generate better recourse for the user.

**Evaluation Details:** By default, we use $\mathcal{D}_{mix}$ distribution to obtain the cost function sample for the EMC objective (i.e $\mathcal{D}_{train} = \mathcal{D}_{mix}$) and the evaluation cost function $\mathcal{C}_u^{\#}$ (i.e $\mathcal{D}_{test} = \mathcal{D}_{mix}$). We use $\mathcal{D}_{mix}$ because it is the most expressive distribution and can combine percentile and step-based costs. Hence, it is more likely to generate samples close to the user's true cost function. See Figure 1 and Section 4.4 to understand why this is important. Note that, this is a significantly better procedure than past works (see §2) which uses a single cost function both during training and evaluation. In our evaluations, the cost samples $\{\mathcal{C}_i\}_{i=1}^M$ used during training to optimize for EMC are different from $\mathcal{C}_u^{\#}$ which is used for evaluation even if they come from the same distribution. Moreover, in Section 5.2;Q3,4,5, we perform distribution shift experiments, where $\mathcal{D}_{train} \neq \mathcal{D}_{test}$ and show that even in such cases INSPIRE generates the best quality recourse.

## 5.2 Research Questions

### Q1. Which Recourse Method Satisfies the Most Users?

In this experiment, we compare different recourse methods on our cost-based evaluation procedure and other indirect metrics. We report the average performance over five random seeds in Table 1. We observe that COLS and P-COLS, which optimize for EMC, achieve 22.64% and 25.89% higher user satisfaction (FS@1) while covering 19.28% and 22.42% more users compared to the strongest baseline on Adult-Income and COMPAS, respectively. We provide results for additional values of $k \in \{0.5, 1, 2, 3\}$ for FS@k in Appendix Table 12. Meanwhile, other methods that optimize for a combination of other indirect objectives, perform worse on user cost-based metrics that directly model user satisfaction. Interestingly, we find that COLS and P-COLS solutions exhibit high feature diversity, proximity, and sparsity. This implies that – (1) the $\mathcal{D}_{mix}$ distribution is generating cost functions that model diverse FPs and COLS along with EMC allowing us to obtain the highest diversity even compared to other methods that directly optimize for it, and (2) proximity, sparsity, and diversity emerge as necessary metrics even under our cost-based evaluation procedure but they are not sufficient to satisfy users with preferences as shown by other methods performance on cost-metrics. This is because it is non-trivial to find the right balance when optimizing for these metrics.

### Q2. Is the Performance Improved by the COLS Optimization Method or the EMC Objective?

We perform ablations to understand the impact of the COLS optimization and the EMC objective. We run a basic local search (LS) to optimize for feature diversity, proximity, and sparsity along with validity. We

Table 3: Robustness to distribution shift. EMC used $\mathcal{D}_{mix}$ to sample cost function for recourse generation while evaluation cost functions are log-percentile shift cost functions from the AR (Ustun et al., 2019).

| Data | Method | FS@1($\uparrow$) | PAC($\downarrow$) | Cov($\uparrow$) |
|------|--------|------------------|-------------------|-----------------|
| **Adult-Income** | **Act. Recourse** | 56.42 | 0.49 | 61.21 |
| | **COLS** | 71.32 | 0.43 | 74.89 |
| | **P-COLS** | **73.68** | **0.42** | **77.11** |
| **COMPAS** | **Act. Recourse** | 70.26 | 0.35 | 76.31 |
| | **COLS** | 80.51 | 0.29 | 81.68 |
| | **P-COLS** | **82.13** | **0.26** | **83.24** |

use a basic local search because there is no efficient way to guarantee reductions in the diversity objective by swapping out single elements from the solution set that is required for using COLS (see Appendix B.1). To quantify the usefulness of COLS, we also optimize EMC using a basic local search.

The results in Table 2 suggest that optimizing for the indirect distance metrics is sub-optimal for user satisfaction. For proximity, sparsity, and feature diversity objectives, the FS score and coverage are very low, while they perform well on their respective metrics. The low FS score for distance metrics is expected as they ignore user preferences and hence can edit features that are not preferred making the generated recourses infeasible under the user's evaluation cost function. We find that EMC with LS outperforms all distance objectives on all cost-based metrics, suggesting that the EMC is a better objective. Meanwhile, the 19% difference in the performance of EMC with LS and COLS can be attributed to our cost optimization (§4.3) that allows COLS to efficiently search the solution space.

## Q3. Robustness to Using Different Distribution for Training and Evaluation Cost Functions

Next, we test the robustness of our method in cases where the evaluation cost functions $\mathcal{C}_u^\#$ for the users are not from the training distribution used in EMC, i.e. $\mathcal{D}_{train} \neq \mathcal{D}_{test}$. We set $\mathcal{D}_{train} = \mathcal{D}_{mix}$ and use *log-percentile shift* cost function from Actionable recourse (Ustun et al., 2019) as the evaluation cost function. The log-percentile shift cost function is defined as $cost(f, x, y) = \log(\frac{1-Q_f(y)}{1-Q_f(x)})$, where $Q_f$ is the CDF of feature $f$ across the population. This leads to a scenario where train and evaluation time cost functions are from a completely different distribution.

In Table 3, we compare with our strongest baseline, Actionable Recourse (AR) (Ustun et al., 2019) which also uses the log-percentile shift cost function during training as an objective. Hence, we observe that the performance of AR improves by 6.49% and 4.46% on Adult-Income and COMPAS datasets respectively when using the log-percentile cost function during evaluation (see Table 1). Moreover, we find that COLS and P-COLS still outperform the AR baseline by 17.26% and 11.87% on Adult-Income and COMPAS datasets respectively. We observe only a slight decrease in the performance of COLS and P-COLS even in this case of a complete distribution shift. This demonstrates the robustness of our method which arises from - (1) This ability of the distribution $\mathcal{D}_{mix}$ to capture a wide variety of plausible cost functions, and (2) The effective exploration of the search space when using COLS with EMC objective.

## Q4. Robustness to Misspecification in cost-type weight

**Design:** Our $\mathcal{D}_{mix}$ distribution samples cost by taking an $\alpha$-weighted combination of linear and percentile costs. These two cost have different underlying assumptions about the how users view the cost of transition between the states. We want to test the robustness of our method in terms of misspecification in users disposition to these types of cost. We perform a robustness analysis where the users cost function has a different $\alpha$ mixing weight as compared to the Monte Carlo samples we use to optimize for EMC. This creates a distribution shift in the user cost function distribution ($\mathcal{D}_{test}$) and the Monte Carlo sampling distribution ($\mathcal{D}_{train}$) used in EMC. We vary the user and Monte Carlo distributions $\alpha$-weights within the range of 0 to 1 in steps of 0.2. At the extremes values of $\alpha = 0, 1$, the shifts are very drastic as the underlying distribution changes completely. In the case when monte carlo $\alpha$ weight is 0 and user $\alpha$ weight is 1 then $\mathcal{D}_{train} = \mathcal{D}_{perc}$ and $\mathcal{D}_{test} = \mathcal{D}_{step}$, similarly for the other case we get $\mathcal{D}_{train} = \mathcal{D}_{step}$ and $\mathcal{D}_{test} = \mathcal{D}_{perc}$. Please note that the distribution $\mathcal{D}_{step}$ and $\mathcal{D}_{perc}$ have completely different underlying principles and are two completely different distributions. Hence, the corners of the heatmap represent drastic distribution shifts.

**Results:** In Figure 2, we show a heatmap plot to which demonstrates the robustness of our method. The color of the block corresponding to Monte Carlo alpha, $\alpha_{mc} = x$ and the users alpha, $\alpha_{user} = y$ represents the

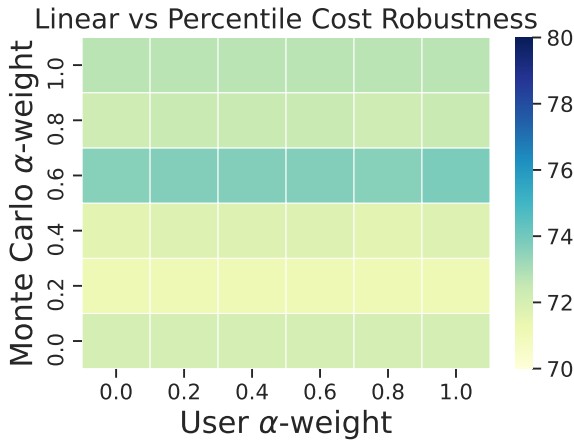

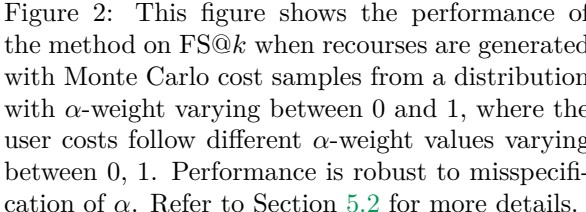

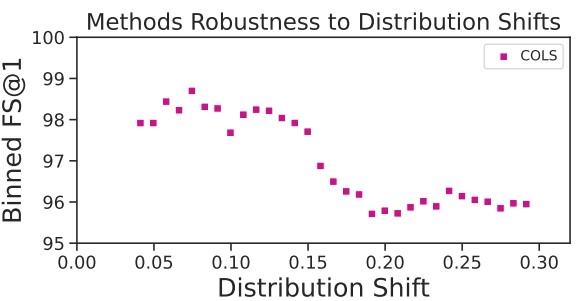

Figure 2: This figure shows the performance of the method on FS@$k$ when recourses are generated with Monte Carlo cost samples from a distribution with $\alpha$-weight varying between 0 and 1, where the user costs follow different $\alpha$-weight values varying between 0, 1. Performance is robust to misspecification of $\alpha$. Refer to Section 5.2 for more details.

Figure 3: In this plot we show the fraction of users satisfied vs the distance between the train and test distributions. The results demonstrate that as the distance increases the performance drops a bit and then plateaus, which means that the method is robust to this kind of distribution shift. Please refer to Section 5.2 for more details.

fraction of users that were satisfied when $\alpha_{mc} = x$ and $\alpha_{user} = y$. This means that if the user thought of costs only in terms of the Linear step involved but the recourse method used samples with only percentile-based cost, still the recourse set can satisfy almost the same number of users. In Figure 3, the corners correspond to these extreme cases described above, the user satisfaction for the top left corner ($\mathcal{D}_{train} = \mathcal{D}_{perc}$ and $\mathcal{D}_{test} = \mathcal{D}_{step}$) is similar to the bottom left corner ($\mathcal{D}_{train} = \mathcal{D}_{step}$ and $\mathcal{D}_{test} = \mathcal{D}_{step}$). Similarly, things happen for the opposite case which is denoted by the top-right ($\mathcal{D}_{train} = \mathcal{D}_{perc}$ and $\mathcal{D}_{test} = \mathcal{D}_{perc}$) and bottom-right ($\mathcal{D}_{train} = \mathcal{D}_{step}$ and $\mathcal{D}_{test} = \mathcal{D}_{perc}$) corners. This means that even when a complete distribution shift occurs the performance user satisfaction remains similar. This can be attributed to the hierarchical step for user preference sampling in the procedure because the preference values can be arbitrary and they scale the raw percentile and linear cost hence the distribution models diverse types of transition costs.

This means that **our methods are robust to misspecification in the train and test distributions.** The almost consistent color of the grid means that **there is very slight variation in the Fraction of Satisfied users when the model is tested on out-of-distribution user cost types.**

### Q5. Are Solutions Robust to Misspecified Feature Scores?

**Design:** In Section 4.1 and Algorithm 4, we define the procedure our distributions use to generate feature scores $\boldsymbol{p}$, assuming random FPs. However, in a population, FPs for users may be clustered in form of subsets of features. For example, to increase *income*, a subgroup of users might prefer to edit their *work hours* whereas another subgroup might prefer a combination of *occupation type* and *education level* as it is easier to change the occupation after attaining a higher degree. Hence, the feature score $\boldsymbol{p}$ for users in a population might be clustered. This is a potential distribution shift in the types of feature scores we generate vs what might exist in real world. Therefore, we consider a scenario where $\mathcal{D}_{train} = \mathcal{D}_{mix}$ but $\mathcal{D}_{test}$ has these clustered feature scores. Hence, the evaluation cost functions $\mathcal{C}_u^{\#}$ are generated from a different distribution.

For users in the Adult-Income data, we use COLS to optimise for EMC using Monte Carlo samples from $\mathcal{D}_{mix}$ (Algorithm 5). To obtain user's evaluation cost functions that differ from this distribution, we first generate 500 different feature subsets indicating which features are editable, where each subset corresponds to a binary vector *concentration* representing a user having specific preferences for some features over others (see Sec. 4.1 and Alg. 4). Since having different editable features induces a different distribution over cost functions, we obtain a measure of distribution shift for each of the 500 *concentration* vectors by taking an $l_2$ distance between the vector and *its nearest neighbor in the space of* concentration *vectors used to generate the recourses.* We use the nearest neighbor because the most outlying *concentration* vectors are least likely to be satisfied by the recourse set. In other words, the likelihood that a user is satisfied depends on the

minimum distance between their *concentration* vector and its nearest neighbor in the cost samples used at recourse generation time. Therefore, when the minimum distance increases, there is a greater distribution shift between the user's cost functions and those obtained from $\mathcal{D}_{mix}$. Finally, we measure how many users are satisfied for a given degree of distribution shift.

**Results:** Figure 3 shows a binned plot of FS@1 against our measure of distribution shift. We observe that as the distance between the distributions increases, the fraction of users satisfied decreases slightly and then plateaus. Even at the maximum distance we obtain, performance has only dropped about 3 points. This implies that **our method is robust to distribution shift in the cost distribution in terms of which features people prefer to edit.** We attribute this to the fact that our $\mathcal{D}_{mix}$ (1) assumes random feature preferences that subsume these skewed preferences and (2) provides multiple recourse options, each of which can cater to different kinds of preferences. Hence, we achieve a good covering of the cost function space.

**Q6. Fairness of Recourse Methods Across Subgroups**

Next, we assess if the recourse methods provide equitable solutions across subgroups based on demographic features like *Gender* and *Race.* This is important because we want to ensure that recourse methods are not further inducing bias towards any particular group because it directly affects the life of users. We adapt existing fairness metrics for disparate impact across population subgroups (Feldman et al., 2015) for the recourse outcomes we study, which we denote by the Disparate Impact Ratio (DIR). Given a metric $\mathcal{M}$, DIR is a ratio between metric scores across two subgroups. DIR-$\mathcal{M} = \mathcal{M}(S=1)/\mathcal{M}(S=0)$. We use either Cov or FS@1 as $\mathcal{M}$. Under the DIR metric, the maximum fairness score that can be achieved

Table 4: Fairness analysis of recourse methods for Gender-based subgroups. **DIR**: Disparate Impact Ratio; **M**: Male, **F**: Female.

| Method | Gender | FS@1 | Cov | DIR-FS | DIR-Cov |
|---|---|---|---|---|---|
| DICE | F | 0.0 | 0.0 | - | - |
| | M | 4.7 | 15.6 | | |
| Face-Eps | F | 12.5 | 22.1 | 1.504 | 1.118 |
| | M | 18.8 | 24.7 | | |
| Face-Knn | F | 29.9 | 36.3 | 0.719 | 0.89 |
| | M | 21.5 | 32.3 | | |
| Act. Recourse | F | 53.8 | 58.7 | 0.881 | 0.959 |
| | M | 47.4 | 56.3 | | |
| Random | F | 7.8 | 34.6 | 0.859 | 0.792 |
| | M | 6.7 | 27.4 | | |
| COLS | F | 72.7 | 76.2 | 0.994 | 0.992 |
| | M | 72.3 | 75.6 | | |
| P-COLS | F | 76.5 | 80.2 | **1.004** | **1.0** |
| | M | 76.8 | 80.2 | | |

is 1, though this might not be achievable depending on the black-box model. We run experiments on the Adult-Income dataset, with a budget of 5000 model queries and $|\mathcal{S}| = 10$.

We present the *gender* and *race* based subgroup results in Table 4 and Appendix Table 6 respectively. We observe that our methods are typically more fair than baselines on both Gender and Race-based subgroups while providing recourse to a larger fraction of people in both subgroups. In particular, we find that our method achieves a score very close to 1 on DIR-FS and DIR-Cov implying a very high degree of fairness. We attribute the fairness of our method to (1) the fact that COLS does not depend on the data distribution, and (2) the use of diverse cost functions to generate recourse.

**Additional Research Questions:** We summarize the additional research questions from Appendix here:

1. We present the computational complexity and runtimes in Appendix A.3.2.
2. Using a larger compute budget improves the performance (Figure 4);
3. We can provide high-quality solutions to the user even with as few as 3 CFs (Figure 5);
4. We can achieve high FS@k with as few as 20 Monte Carlo samples (Figure 7);
5. Our method works for other black-box classification models as well (Table 11);
6. We also present some qualitative examples of recourses provided by our method in Table 8.

## 6 Discussion and Conclusion

Our novel method INSPIRE provides a way to incorporate FPs in the recourse generation and evaluation process. INSPIRE lays a foundation for future works to build more complex distributions to better represent the population by designing non-linear transition costs or modifying the COLS procedure to account for the causal relationships between features while accounting for individual user preferences. We show that our method achieves much higher rates of user satisfaction than comparable baselines and is robust to distribution shifts. Additionally, we provide an Ethics and Reproducibility statement in Appendix A.1 and A.2.

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

# A Appendix for INSPIRE: Incorporating Diverse Feature Preferences in Recourse

## A.1 Ethics Statement

We hope that our recourse method is adopted by institutions seeking to provide reasonable paths to users for achieving more favorable outcomes under the decisions of black-box machine learning models or other inscrutable models. We see this as a "robust good," similar to past commentators Venkatasubramanian & Alfano (2020). Below, we comment on a few other ethical aspects of the algorithmic recourse problem.

First, we suggest that fairness is an important value that recourse methods should always be evaluated along, but we note that evaluations will depend heavily on the model, training algorithm, and training data. For instance, a sufficiently biased model might not even allow for suitable recourses for certain subgroups. As a result, any recourse method will fail to identify an equitable set of solutions for the population. That said, recourse methods can still be designed to be more or less fair. This much is evident from our varying results on fairness metrics using a number of recourse methods. What will be valuable in future work is to design experiments that separate the effects on the fairness of the model, training algorithm, training data, and recourse algorithm. Until then, we risk blaming the recourse algorithm for the bias of a model, or vice versa.

Additionally, there are possible dual-use risks from developing stronger recourse methods. For instance, malicious actors may use recourse methods when developing models in order to *exclude* certain groups from having available recourse, which is essentially a reversal of the objective of training models for which recourse is guaranteed (Ross et al., 2021). We view this use case as generally unlikely, but pernicious outcomes are possible. We also note that these kinds of outcomes may be difficult to detect, and actors may make bad-faith arguments about the fairness of their deployed models based on other notions of fairness (like whether or not a model has access to protected demographic features) that distract from an underlying problem in the fairness of recourses.

## A.2 Reproducibility Statement

To encourage reproducibility, we provide our source code, including all the data pre-processing, model training, recourse generation, and evaluation metric scripts as supplementary material. The details about

Table 5: Table containing data statistics and black-box model details. The binary version of the datasets are take from (Pawelczyk et al., 2021) whereas the non-binary version are taken from (Mothilal et al., 2020).

|  | Adult-Income Binary | COMPAS Binary | Adult-Income | COMPAS |
|---|---|---|---|---|
| # Continuous features | 3 | 4 | 2 | 3 |
| # Categorical features | 9 | 3 | 10 | 12 |
| Undesired class | $\leq$ 50k | Will Recidivate | $\leq$ 50k | Will Recidivate |
| Desired class | $>$ 50k | Won't Recidivate | $>$ 50k | Won't Recidivate |
| Train/val/test | 20088/2338/749 | 1415/229/491 | 13172/1569/748 | 5491/705/444 |
| Model Type | ANN(2, 20) | ANN(2, 20) | ANN(2, 20) | ANN(2, 20) |
| Val Accuracy | 82% | 69% | 81% | 61% |

the datasets and the pre-processing are provided in Appendix A.3.1. We also provide clear and concise Algorithms 5, 2, 3 for our cost sampling procedures and our optimization method COLS in Algorithm 1. Additionally, we also provide formal proof of the Theorem 4.1 stated in the main paper in Appendix B.2.2 along with the constructive procedure for the proof provided in Algorithm 1.

## A.3 Experimental Setup

### A.3.1 Datasets and Black-Box Model

In our experiments, we have two versions of the dataset, one with binary categorical features, whereas the other with non-binary categorical features. In the main paper, we show results on the binarized version (Table 1) as an important baseline, Actionable Recourse (Ustun et al., 2019), operates with binary categorical features.[3] The data statistics for all the datasets can be found in Table 5. In our experiments, for all the datasets, the features gender and race are considered to be immutable (Mothilal et al., 2020), since we perform subgroup analysis with these variables that would be rendered meaningless if users could switch subgroups. Other features can either be mutable or conditionally mutable depending on semantics. These constraints can be incorporated into the methods by providing a schema of feature mutability criterion. Our black-box model is a multi-layer perceptron model with 2 hidden layers trained on the trained set and validated on the dev set. The accuracy numbers are shown in Table 5. The test set which is used in the counterfactual generation experiments only contains users which are classified to the undesired class by the trained black-box model. Note that our method can operate with any type of model, the only requirement is the ability to query the model for outcome given a user's state vector.

### A.3.2 Computational Complexity:

COLS is a local search-based method and runs for $\mathcal{O}(\frac{B}{|S|})$ iterations for each user to generate the recourse set, where B is the budget (see section 5.1 - Baselines). The complexity of the cost optimization step in COLS is $\mathcal{O}(|S|^2 * M)$ per iteration. Values of $|S|$ and M as low as 3 and 10 respectively work well in practice (see Appendix B.2 and Figure 5, 7). Finally for the current implementation the wall clock time on the adult dataset for each user with $|S| = 10$, M = 100, B = 5000 setting is COLS = 20s, Random = 7.5s, DICE = 7.5s, AR = 11s, Face-knn = 7s, Face-Eps = 6s (can be parallelized across users). Cost function samples can be pre-computed once and saved for all experiments, this typically takes a few minutes ($<$ 5 min) across all users.

### A.3.3 Recourse Generation and Evaluation Pipeline

To approximate the expectation in equation 3, our algorithm samples multiple cost functions $\{\mathcal{C}_i\}_{i=1}^M \sim \mathcal{D}_{train}$, which are used in EMC to generate the recourse set for the user. In the generation phase, we use Equation 4 as our objective. Note that, this objective promotes that the generated recourse set contains at least one good CF for each of the cost samples, hence this set satisfies a large variety of samples from $\mathcal{D}_{train}$. This is achieved via minimizing the mean of the minimum cost incurred for each of the Monte Carlo samples

---

[3]The binary datasets can be downloaded from https://github.com/carla-recourse/cf-data, whereas the non-binary data can be found on https://github.com/interpretml/DiCE.

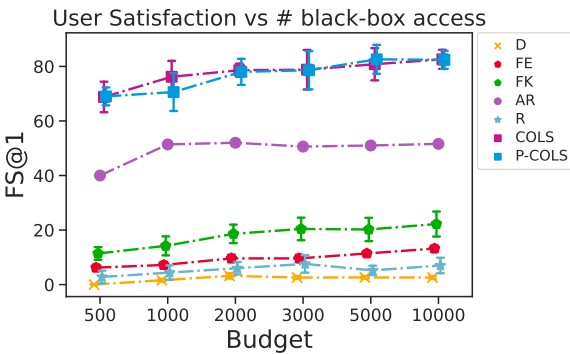

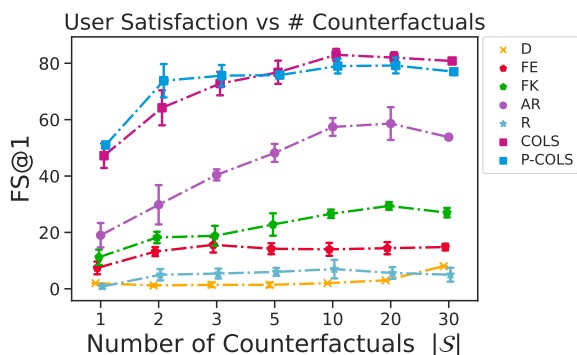

Figure 4: Figure showing the performance of different recourse methods as the Budget is increased. These are the average number across 5 different runs along with the standard deviation error bars. For some methods the standard deviation is very low hence not visible as bars in the plot. It can be seen that as the budget increases the performance of COLS and P-COLS increases. Please refer to Section A.4 for more details.

Figure 5: Figure showing the performance of different recourse methods as the the number of counterfacuals to be generated is increased. These are the average number across 5 different runs along with standard deviation error bars. We see that there is a monotonic increase in the fraction of users satisfied as the size of the set increases. We also observe that most of the performance can be obtained with a small set size. Please refer to Section A.4 for more details.

(Robert & Casella, 2010). Equivalently, the objective is minimized by a set of counterfactuals $\mathcal{S}$ where for each cost function there exists an element in $\mathcal{S}$ which incurs the least possible cost. In practice the size of set $\mathcal{S}$ is restricted, hence we may not achieve the absolute minimum cost but the objective tries to ensure that the counterfactuals which belong to the set have a low cost at least with respect to one Monte Carlo cost sample. The generation phase outputs a set of counterfactuals $\mathcal{S}$ which is to be provided to the users as recourse options. Given this set $\mathcal{S}_u$, in the evaluation phase, we use the user's simulated evaluation cost functions which are not available in the generation phase to compare different methods, to compute the cost incurred by the user $\mathrm{MinCost}(\boldsymbol{s}_u, \mathcal{S}; \mathcal{C}_u^{\#})$ and calculate the metrics defined in the Section 5.1.

## A.4 Additional Research Questions

### Q7. Does Method Performance Scale with Available Compute?

**Design:** In this experiment on the Adult-Income dataset, we measure the change in performance of all the models as the number of access to the black-box model (budget) increases. Ideally, a good recourse method should be able to exploit these extra queries and use it to satisfy more users. We vary the allocated budget in the set $\{500, 1000, 2000, 3000, 5000, 10000\}$ and report the FS@1. We run the experiment on a random subset of 100 users for 5 independent runs and then report the average performance with standard deviation-based error bars in Figure 4.

**Results:** In Figure 4, we can see that **as the allocated budget increases the performance of the COLS and P-COLS increases** and then saturates. This suggests that our method can exploit the additional black-box access to improve the performance. Other methods like AR and Face-Knn also show performance improvement but our method COLS and P-COLS consistently upper-bound their performance. **Our method satisfies approximately 70% of the user with a small budget of 500** and quickly starts to saturate around a budget of 1000. This suggests that **our methods are suitable even under tight budget constraints** as they can achieve good performance rapidly. For example, in a real-world scenario where the recourse method is deployed and has to cater to a large population, in such cases there might be budget constraints imposed onto the method where achieving good quality solution quickly is required. Lastly, for DICE and Random search the performance on the FS@1 increase by a very small margin and

Table 6: Fairness analysis of recourse methods for subgroups with respect to Race. **DIR**: Disparate Impact Ratio; **W**: White, **NW**: Non-White (Section 5.2).

| Method | Race | FS@1 | Cov | DIR-FS | DIR-Cov |
|---|---|---|---|---|---|
| DICE | NW | 0.0 | 0.0 | - | - |
| | W | 3.1 | 10.4 | | |
| Face-Eps | NW | 7.7 | 12.7 | 2.312 | 2.047 |
| | W | 17.8 | 26.0 | | |
| Face-Knn | NW | 12.7 | 25.4 | 2.228 | 1.425 |
| | W | 28.3 | 36.2 | | |
| Act. Recourse | NW | 46.5 | 54.9 | 1.101 | **1.056** |
| | W | 51.2 | 58.0 | | |
| Random | NW | 4.9 | 28.9 | 1.571 | 1.076 |
| | W | 7.7 | 31.1 | | |
| COLS | NW | 67.6 | 71.1 | 1.089 | 1.082 |
| | W | 73.6 | 76.9 | | |
| P-COLS | NW | 72.5 | 74.6 | **1.07** | 1.092 |
| | W | 77.6 | 81.5 | | |

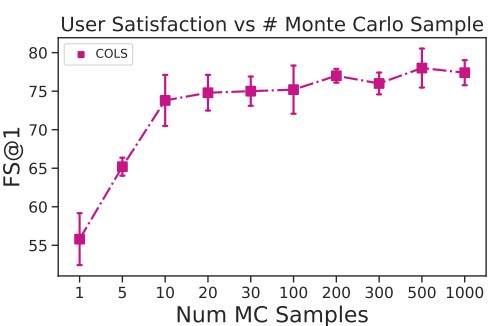

Table 7: Figure showing the performance of the COLS method as the number of Monte Carlo samples increase. These are the average number across 5 different runs along with standard deviation error bars. There is a steep increase and then the performance saturates. This implies that in practice we do not need a large number of samples to converge to the higher user satisfaction. Refer to Section A.4 for more details.

then stays constant as these methods are trying to optimize for different objectives which don't align well with user satisfaction as demonstrated in Section 5.2.

**Q8. Does providing more options to users help?**

**Design:** In this experiment, we measure the effect of having flexibility to provide the user with more options, i.e. a bigger set $\mathcal{S}$. The question here is that can the methods effectively exploit this advantage and provide lower cost solution sets to the user such that the overall user satisfaction is improved. In this experiment on the Adult-Income dataset, we take a random subset of 100 users and fix the budget to 5000, Monte Carlo cost sample is set to 1000 and then vary the size of the set $\mathcal{S}$ in the set $\{1, 2, 3, 5, 10, 20, 30\}$. We restrict the size of the set to a maximum of 30 as beyond a point it becomes hard for users to evaluate all the recourse options and decide which one to act upon. We run 5 independent runs for all the data points and plot the mean performance along with standard deviation error bars. In Figure 5, we plot the fraction of users satisfied @1 as the size of set $\mathcal{S}$ is increased.

**Result:** We observed that **COLS and P-COLS monotonically increase the FS@1 metric as $|\mathcal{S}|$ increases** from 1 to 30. This is consistent with the intuition behind our methods (See Figure 1, section 4.2, A.3.3 for more details). It is a fundamental property of our objective that as $|\mathcal{S}|$ increases towards $M$ which is 1000 in this case, then the quality of the solution set should increase and reach the best possible value that can be provided under the user's cost function. We note empirically that **smaller set size $|\mathcal{S}|$ between 3 to 10 is enough in most practical cases** to reach close to maximum performance. Additionally, even with $|\mathcal{S}| \in \{1, 2, 3\}$ our methods significantly outperform all the other methods in terms of the number of users satisfied. This property is useful in real-world scenarios where the deployed recourse method can provide as little as 3 options while still satisfying a large fraction of users. Additionally, we also see improvement in the case of AR and Face-Knn methods as $|\mathcal{S}|$ increases. Note that Randoms Search's performance doesn't change as we increase the set size because the method doesn't take local steps from the best set and samples random points from a very large space, hence it is much harder to end up with low-cost counterfactuals.

**Q9. Does increase the number of Monte Carlo samples help with user satisfaction?**

**Design:** In this experiment, we want to demonstrate the effect of increasing the number of Monte Carlo samples on the performance of our COLS method. We take a random subset of 100 users, a budget of 5000, $|\mathcal{S}| = 10$. We vary the number of Monte Carlo samples (M) in the set $\{1, 5, 10, 20, 30, 100, 200, 300, 500, 1000\}$

Table 8: Table providing qualitative examples for two users from the dataset. We show each users state vector, the features that user is willing to edit, the preference scores for those editable features, the recourses provided and the cost of the generated recourses. In the first example we see that user highly prefers the feature *capital loss* and the recourse which suggests edit to that has the lowest cost for the user. Whereas, the recourse which makes changes to both *Occupation* and *Capital Loss* has the highest cost as its changing multiple features. For the second user, we see that the most preferred feature is *Education-Num* but the changes suggested in the recourse requires three steps 7-8-9-10, hence the cost for that recourse is not the lowest but still relatively low. Whereas, the recourse suggesting smaller changes to *Capital Loss* which is the second most preferred feature has the lowest cost for the user.

| Feature Name | State Vector | Editable Features | Preference scores | Recourses | Cost |
|---|---|---|---|---|---|
| Age | 24 | No | 0 | | |
| Workclass | Private | No | 0 | $\left(\text{Capital Loss: } 0 \to 1\right)$ | 0.009 |
| Education-Num | 10 | No | 0 | | |
| Martial-Status | Married | No | 0 | | |
| Occupation | Other | Yes | 0.055 | | |
| Relationship | Husband | No | 0 | $\left(\text{Occupation: Other} \to \text{Manager}\right)$ | 0.378 |
| Race | White | No | 0 | | |
| Gender | Male | No | 0 | | |
| Capital Gain | 0 | No | 0 | | |
| Capital Loss | 0 | Yes | 0.944 | $\left(\begin{array}{l}\text{Occupation: Other} \to \text{Manager} \\ \text{Capital Loss: } 0 \to 1\end{array}\right)$ | 0.387 |
| # Work Hours | 40 | No | 0 | | |
| Country | US | No | 0 | | |
| Age | 45 | No | 0 | | |
| Workclass | Private | No | 0 | $\left(\text{Capital Loss: } 0 \to 1\right)$ | 0.071 |
| Education-Num | 7 | Yes | 0.537 | | |
| Martial-Status | Married | No | 0 | | |
| Occupation | Other | No | 0 | $\left(\text{Capital Gain: } 0 \to 1\right)$ | 0.106 |
| Relationship | Non-Husband | No | 0 | | |
| Race | White | No | 0 | | |
| Gender | Female | No | 0 | $\left(\text{Education-Num: } 7 \to 10\right)$ | 0.187 |
| Capital Gain | 0 | Yes | 0.078 | | |
| Capital Loss | 0 | Yes | 0.240 | | |
| # Work Hours | 32 | Yes | 0.142 | $\left(\text{\# Work Hours: } 32 \to 70\right)$ | 0.695 |
| Country | US | No | 0 | | |

and compute the user satisfaction. We ran 5 different runs with different Monte Carlo samples and show the average FS@1 along with the standard deviation in the Figure 7.

**Results:** We observe that **as the number of Monte Carlo samples increases, the performance of the method on the FS@1 metric monotonically increases.** This supports the intuition underlying our method (see Figure 1). That is, given a user with a cost function $\mathcal{C}_u^*$ as we get more and more samples from the cost distribution $\mathcal{D}_{train}$ the probability of having a cost sample similar to $\mathcal{C}_u^*$ increases and hence the fraction of satisfied users increase. It is important to note that **empirically the method's performance approaches maximum user satisfaction with as low as 20 Monte Carlo samples.** In real-world scenarios, where the deployed model is catering to a large population this can lead to small recourse generation time, hence making it more practical.

**Q10. Qualitative examples of the recourses generated for some of the users.**

In Table 8, we show a few examples of users along with their state vector, their editable features, their preference scores along with the recourses provided to them and their cost.

**Q11. Comparison of methods on Non Binary Dataset?**

In Table 9, we show the results on the non-binary version of the dataset. We observe similar performance on and trends in these results as well. COLS and P-COLS performs the best in terms of user satisfaction.

**Q12. Robustness to black-box model architecture families and randomness?**

In this experiment we demonstrate the result of our model when we train the same ANN architecture with different random seed (Table 10) and when we change the model family to a logistic regression classifier (Table 11). These obtained results have similar trends and demonstrate the effectiveness and robustness of

Table 9: Table comparing different recourse methods across various cost and distance metrics on Non-Binary versions of the datasets (Section A.3.1).The numbers reported are averaged across 5 different runs. Variance values have been as 89% of them were lower than 0.05, with the maximum being 0.86. FS@1: Fraction of users satisfied at $k = 1$. PAC: Population Average Cost. Cov: Population Coverage. For all the metrics higher is better except for PAC where lower is better.

| Data | Method | Metrics | | | | | | |
|------|--------|---------|---|---|---|---|---|---|
| | | Cost Metrics | | | Distance Metrics | | | |
| | | FS@1 | PAC($\downarrow$) | Cov | Div | Prox | Spars | Val |
| Adult-Income - NB | DICE | 6.28 | 1.45 | 27.01 | 53.01 | 57.02 | 47.80 | 86.20 |
| | Random | 0.08 | 2.42 | 17.41 | **70.35** | 33.32 | 22.45 | 75.71 |
| | COLS | 72.67 | **0.36** | **74.60** | 29.27 | **79.06** | **76.64** | **97.85** |
| | P-COLS | **70.03** | 0.39 | 72.81 | 29.85 | 78.45 | 76.29 | 92.30 |
| COMPAS - NB | DICE | 14.86 | 1.02 | 25.45 | 27.88 | 82.38 | 69.44 | **99.86** |
| | Random | 1.31 | 1.87 | 21.76 | **49.07** | 54.10 | 42.34 | 67.82 |
| | COLS | 67.34 | **0.31** | 68.11 | 20.53 | 85.47 | 82.34 | 95.97 |
| | P-COLS | **70.86** | 0.35 | **72.03** | 21.03 | **85.48** | **82.88** | 91.93 |

Table 10: Table comparing different recourse methods across various cost and distance metrics for a black-box model with different seed but belonging to the same model family. The numbers reported are averaged across 5 different runs.

| Data | Method | Metrics | | | | | | |
|------|--------|---------|---|---|---|---|---|---|
| | | Cost Metrics | | | Distance Metrics | | | |
| | | FS@1 | PAC($\downarrow$) | Cov | Div | Prox | Spars | Val |
| Adult-Income | DICE | 2.70 | 1.24 | 7.10 | 3.80 | 66.20 | 47.30 | 97.80 |
| | Face-Eps | 13.32 | 0.79 | 19.88 | 5.43 | 91.97 | 74.80 | **100.00** |
| | Face-Knn | 21.78 | 0.83 | 34.13 | 8.67 | 88.68 | 71.43 | **100.00** |
| | Act. Recourse | 46.55 | 0.58 | 53.82 | 19.07 | 74.33 | 73.25 | 80.72 |
| | Random | 5.71 | 1.42 | 28.24 | **48.93** | 55.10 | 39.30 | 78.73 |
| | COLS | 75.12 | **0.36** | 77.40 | 25.43 | 81.00 | 77.70 | 98.28 |
| | P-COLS | **75.76** | 0.38 | **79.14** | 25.54 | **81.84** | **78.38** | 95.10 |
| COMPAS | DICE | 0.90 | 0.88 | 1.50 | 12.50 | 63.90 | 30.70 | 99.30 |
| | Face-Eps | 6.80 | 0.29 | 6.80 | 2.40 | **95.00** | 60.40 | **100.00** |
| | Face-Knn | 6.80 | 0.29 | 6.80 | 2.40 | 94.90 | 60.30 | **100.00** |
| | Act. Recourse | 56.24 | 0.45 | 58.48 | 9.72 | 80.12 | **73.62** | 39.10 |
| | Random | 27.44 | 0.78 | 35.70 | **41.76** | 58.14 | 33.06 | 49.34 |
| | COLS | 77.08 | **0.24** | 77.90 | 29.33 | 76.90 | 68.87 | 95.78 |
| | P-COLS | **78.32** | **0.24** | **79.02** | 29.02 | 77.88 | 70.08 | 92.10 |

our methods COLS and P-COLS which consistently satisfy cover and satisfy more users with low average population costs. In Table 10, we show the results when we train another black-box model with a different seed to see the effect of having a different trained model from the same model family.

**Q13. Additional results for different values of $k$ in FS@$k$**
 In Table 12, we report the fraction of satisfied user metric FS@$k$ for four different values of $k \in \{0.5, 1, 2, 3\}$. These results are an extension of the results presented in Table 1.

# B  Appendix - Objective and Optimization

## B.1  Proposed Method

### B.1.1  Other Objectives

To obtain feasible a counterfactual set, past works have used various objective terms. We list objectives below from methods we compare with.

**1. DICE** (Mothilal et al., 2020) optimizes for a combination of Distance Metrics like *diversity* and *proximity*. They model diversity via Determinantal Point Processes (Kulesza & Taskar, 2012) adopted for solving subset

Table 11: Table comparing different recourse methods across various cost and distance metrics for a logistic regression black-box model. The numbers reported are averaged across 5 different runs.

| Data | Method | Metrics | | | | | | |
|------|--------|---------|---|---|---|---|---|---|
| | | Cost Metrics | | | Distance Metrics | | | |
| | | FS@1 | PAC($\downarrow$) | Cov | Div | Prox | Spars | Val |
| Adult-Income | D | 1.30 | 1.47 | 7.10 | 6.50 | 64.80 | 49.20 | 76.60 |
| | FE | 2.82 | 0.99 | 5.46 | 9.44 | 83.08 | 65.90 | **100.00** |
| | FK | 17.04 | 0.90 | 28.08 | 7.22 | 83.98 | 67.80 | **100.00** |
| | AR | 44.40 | 0.62 | 52.62 | 21.74 | 74.00 | 72.92 | 87.58 |
| | R | 3.60 | 1.59 | 24.10 | **48.14** | 54.76 | 38.94 | 79.46 |
| | COLS | 67.93 | **0.39** | 69.97 | 27.83 | 78.40 | 74.43 | 99.13 |
| | P-COLS | **69.17** | 0.40 | **71.57** | 27.20 | **79.30** | **76.70** | 95.67 |
| COMPAS | D | 0.00 | - | 0.00 | 11.10 | 63.10 | 29.20 | 100.00 |
| | FE | 6.30 | **0.16** | 6.30 | 3.60 | **95.00** | 60.50 | **100.00** |
| | FK | 6.30 | **0.16** | 6.30 | 3.60 | **95.00** | 60.50 | **100.00** |
| | AR | 74.32 | 0.31 | 74.32 | 15.66 | 80.98 | 74.26 | 53.66 |
| | R | 28.76 | 0.77 | 36.96 | **43.22** | 56.22 | 32.00 | 82.10 |
| | COLS | 87.88 | 0.18 | 87.88 | 31.92 | 76.93 | 71.63 | 89.33 |
| | P-COLS | **89.25** | 0.17 | **89.25** | 28.17 | 81.13 | **74.73** | 91.62 |

Table 12: Cost metrics for additional $k$ values in FS@k for the results presented in the main Table 1.

| Data | Method | Cost Metrics | | | |
|------|--------|--------------|---|---|---|
| | | FS@0.5 | FS@1 | FS@2 | FS@3 |
| Adult-Income | DICE | 0.4 | 2.47 | 6.9 | 8.23 |
| | Face-Eps | 7.32 | 15.23 | 22.57 | 22.6 |
| | Face-Knn | 11.08 | 25.3 | 34.82 | 35 |
| | Act. Recourse | 28.03 | 49.93 | 56.78 | 56.85 |
| | Random | 0.82 | 6.27 | 28.5 | 31.68 |
| | COLS | 53.4 | 72.57 | 76.05 | 76.07 |
| | P-COLS | **53.42** | **75.82** | **79.18** | **79.2** |
| COMPAS | DICE | 0.2 | 0.4 | 0.4 | 0.4 |
| | Face-Eps | 10.4 | 12.2 | 12.2 | 12.2 |
| | Face-Knn | 10.4 | 12.2 | 12.2 | 12.2 |
| | Act. Recourse | 42.47 | 65.8 | 66.6 | 66.6 |
| | Random | 7.85 | 29.95 | 39.2 | 39.2 |
| | COLS | 73.07 | 82.23 | 82.23 | 82.23 |
| | P-COLS | **74.33** | **83.73** | **83.73** | **83.73** |

selection problems with diversity constraints. They use determinant of the kernel matrix given by the counterfactuals as their diversity objective as defined below.

$$dpp\_diversity(\mathcal{S}) = det(\mathbf{K}), \text{ where} \mathbf{K}_{ij} = \frac{1}{1 + dist(\boldsymbol{s}_i, \boldsymbol{s}_j)}$$

Here, $dist(\boldsymbol{s}_i, \boldsymbol{s}_j)$ is the normalized distance metric as defined in Wachter et al. (2017) between two state vectors. *Proximity* is defined in terms of the distance between the original state vector and the counterfacutals, $prox(\boldsymbol{x}, \mathcal{S}) = 1 - \frac{1}{N} \sum_{i=1}^{|\mathcal{S}|} dist(\boldsymbol{x}, \mathcal{S}_i)$, where $\mathcal{S}_i$ is a counterfactual.

**2. Actionable Recourse** (Ustun et al., 2019) work under the assumption that all features have equal preference scores for all the users. They define cost function based on the log-percentile shift is given by,

$$cost(\boldsymbol{s} + a; \boldsymbol{s}) = \sum_{j \in \mathcal{J}_A} \log \frac{1 - Q_j(\boldsymbol{s}_j + a_j)}{1 - Q_j(\boldsymbol{s}_j)}$$

---

**Algorithm 2** Sampling Cost Functions from $\mathcal{D}_{perc}$

---

**Input:** State vector $\boldsymbol{s}$, Feature Scores $\boldsymbol{p}$
**Output:** Percentile based Transition Cost functions $\mathcal{C}$.
**function** PerCost($\boldsymbol{s}, \boldsymbol{p} = \text{None}$)
    **forall** $f_i \in \mathcal{F}$ **do**
        `// ` $s_i$ `value of feature` $f_i$ `in s.`
        **if** $\boldsymbol{p}^{f_i} = 0$ **then**
            $\mathcal{C}(f_i, s_i, .) = \infty$
            $\mathcal{C}(f_i, s_i, s_i) = 0$
        **else**
            **if** $f_i$ *is ordered* **then**
                **if** $f_i$ *can only increase* **then**

$$\mathcal{C}(f_i, s_i, x) = \begin{cases} |getPercentile(x) - getPercentile(s_i)| & \forall x > s_i \\ 0 & \forall x = s_i \\ \infty & \forall x < s_i \end{cases}$$

                **else if** $f_i$ *can only decrease* **then**

$$\mathcal{C}(f_i, s_i, x) = \begin{cases} |getPercentile(s_i) - getPercentile(x)| & \forall x < s_i \\ 0 & \forall x = s_i \\ \infty & \forall x > s_i \end{cases}$$

                **else if** $f_i$ *can both increase or decrease* **then**

$$\mathcal{C}(f_i, s_i, x) = \begin{cases} |getPercentile(x) - getPercentile(s_i)| & \forall x > s_i \\ 0 & \forall x = s_i \\ |getPercentile(s_i) - getPercentile(x)| & \forall x < s_i \end{cases}$$

            **else if** $f_i$ *is unordered* **then**
                $\mathcal{C}(f_i, s_i, .) = Uniform(0, 1)$
        **if** $\boldsymbol{p}$ *is not None* **then**
            $\mathcal{C}(f_i, s_i, .) \leftarrow \mathcal{C}(f_i, s_i, .) * (1 - p^{f_i})$
    **end**
    **return** $\mathcal{C}$
**end**

---

where $Q_j(.)$ is the cumulative distribution function of $\boldsymbol{s}_j$ in the target population, $\mathcal{J}_A$ is the set of actionable features and $a_j$ is the action performed on the feature $j$.

## B.2 Optimization Methods

**Notation:** We assume that we have a dataset with features $\mathcal{F} = \{f_1, f_2, ... f_k\}$. Each feature can either be continuous $\mathcal{F}^{con} \subset \mathcal{F}$ or categorical $\mathcal{F}^{cat} \subset \mathcal{F}$. Each continuous feature $f_i^{con}$ takes values in the range $[r_i^{min}, r_i^{max}]$, which we discretize to integer values. For a continuous feature $f_i$, we define the range $Q^{(f_i)} = \{k \in \mathbb{Z} : k \in [r_i^{min}, r_i^{max}]\}$ and for a categorical feature $f_i$, we define it as $Q^{(f_i)} = \{q_1^{f_i}, q_2^{f_i}, ..., q_{d_i}^{f_i}\}$, where $q_{(.)}^{f_i}$ are the states that feature $f_i$ can take. Features can either be mutable ($\mathcal{F}^m$), conditionally mutable ($\mathcal{F}^{cm}$), or immutable ($\mathcal{F}^{\oslash}$), according to the real-world causal processes that generate the data. Mutable features can transition from between any pair of states in $Q^{(f_i)}$; conditionally mutable features can transition between pairs of states only when permitted by certain conditions; and immutable features cannot be changed under any circumstances. For example, *Race* is an immutable feature (Mothilal et al., 2020), *Age* and *Education* are conditionally mutable (cannot be decreased under any circumstances), and *number of work hours* is mutable (can both increase and decrease). Lastly, while continuous features inherently define an ordering in its values, categorical features can either be ordered or unordered based on its semantic meaning. For instance, *Age* is an ordered feature that is conditionally mutable (can only increase).

---

**Algorithm 3** Sampling Cost Functions from $\mathcal{D}_{step}$

---

**Input:** State vector $\boldsymbol{s}$, Feature Scores $\boldsymbol{p}$
**Output:** Number of Steps based Transition Cost functions $\mathcal{C}$.
**function** StepCost($\boldsymbol{s}, \boldsymbol{p} = $ None)

$\quad$ **for** $f_i \in \mathcal{F}$ **do**
$\quad\quad$ **if** $\boldsymbol{p}^{f_i} = 0$ **then**
$\quad\quad\quad$ $\mathcal{C}(f_i, s_i, .) = \infty$
$\quad\quad\quad$ $\mathcal{C}(f_i, s_i, s_i) = 0$
$\quad\quad$ **else**
$\quad\quad\quad$ **if** $f_i$ *is ordered* **then**
$\quad\quad\quad\quad$ **if** $f_i$ *can only increase* **then**

$$\mathcal{C}(f_i, s_i, x) = \begin{cases} \frac{|\{y \mid y > s_i \wedge y \leq x\}|}{|\{y \mid y > s_i\}|} & \forall x > s_i \\ 0 & \forall x = s_i \\ \infty & \forall x < s_i \end{cases}$$

$\quad\quad\quad\quad$ **else if** $f_i$ *can only decrease* **then**

$$\mathcal{C}(f_i, s_i, x) = \begin{cases} \frac{|\{y \mid y < s_i \wedge y \geq x\}|}{|\{y \mid y < s_i\}|} & \forall x < s_i \\ 0 & \forall x = s_i \\ \infty & \forall x > s_i \end{cases}$$

$\quad\quad\quad\quad$ **else if** $f_i$ *can both increase or decrease* **then**

$$\mathcal{C}(f_i, s_i, x) = \begin{cases} \frac{|\{y \mid y > s_i \wedge y \leq x\}|}{|\{y \mid y > s_i\}|} & \forall x > s_i \\ 0 & \forall x = s_i \\ \frac{|\{y \mid y < s_i \wedge y \geq x\}|}{|\{y \mid y < s_i\}|} & \forall x < s_i \end{cases}$$

$\quad\quad\quad$ **else if** $f_i$ *is unordered* **then**
$\quad\quad\quad\quad$ $\mathcal{C}(f_i, s_i, .) = Uniform(0, 1)$
$\quad\quad$ **if** $\boldsymbol{p}$ *is not None* **then**
$\quad\quad\quad$ $\mathcal{C}(f_i, s_i, .) \leftarrow \mathcal{C}(f_i, s_i, .) * (1 - p^{f_i})$
$\quad$ **end**
$\quad$ **return** $\mathcal{C}$
**end**

---

**Algorithm 4** Sampling Random Feature Scores.

---

**Output:** Feature Scores $\boldsymbol{p}$.
**function** sampleFeatureScores($\mathcal{F}_p = \{\}, \boldsymbol{p} = None$)

$\quad$ **if** $\mathcal{F}_p$ *is {}* **then**
$\quad\quad$ $\mathcal{F}_p \sim RandomSubset(\mathcal{F}_{mutable})$

$\quad$ **if** $\boldsymbol{p}$ *is None* **then**
$\quad\quad$ $concentration = [1 \; if \; f \in \mathcal{F}_p \; else \; 0 \; for \; f \; in \; \mathcal{F}]$
$\quad\quad$ $\boldsymbol{p} \sim Dirichlet(concentration)$

$\quad$ **return** $\boldsymbol{p}$
**end**

---

### B.2.1 Hierarchical Cost Sampling Procedure

To optimize for EMC, we need a plausible distribution which can model users' cost functions. We propose a hierarchical cost sampling distribution which provides cost samples that are a linear combination of *percentile shift cost* (Ustun et al., 2019) and *linear cost*, where the weights of this combination are user-specific. *Percentile shift cost* for ordered features is proportional to the change in a feature's percentile associated with the change from an old feature value to a new one. E.g., if a user is asked to increase the number of work hours from 40 to 70, then given the whole dataset, we can estimate the percentile of users working 40 and 70 hours a week. The cost incurred is then proportional to the difference in these percentiles. The *Linear cost* for ordered features is proportional to the number of intermediate states a user will have to go through while transitioning from their current state to the final state. E.g., if a user is asked to change their

---

**Algorithm 5** Sampling Cost Functions from $\mathcal{D}_{mix}$.

---

**Input:** State vector $\boldsymbol{s}$, feature scores $\boldsymbol{p}$, *Optional:* cost-type mixing weight $\alpha$
**Output:** Cost functions $\mathcal{C}$.
**function** sampleCost($\boldsymbol{s}, \boldsymbol{p}, \alpha = $ None)
    **if** $\alpha$ *is None* **then**
        |   $\alpha \sim Uniform(0,1)$

    ▷ Get Step and Percentile cost.
    $\mathcal{C}^{(Lin)} \leftarrow LinCost(\boldsymbol{s}, \boldsymbol{p})$
    $\mathcal{C}^{(Perc)} \leftarrow PerCost(\boldsymbol{s}, \boldsymbol{p})$
    $\mathcal{C}^{(Mix)} \longleftarrow \alpha * \mathcal{C}^{(Lin)} + (1 - \alpha) * \mathcal{C}^{(Perc)}$
    **return** $\mathcal{C}^{(Mix)}$
**end**

---

**Algorithm 6** Algorithm for Theorem 4.1

---

**Input:** $\mathbf{C}^b, \mathbf{C} \in \mathbb{R}^{N \times M}$ matrices containing the costs with respect to all cost samples..
**Output:** $\mathbf{B} \in \mathbb{R}^{N \times N}$, matrix containing the benefits of replacing pairs from $\mathcal{S}_{t-1}^{best} \times \mathcal{S}_t$
**function** computeBenefits($\mathbf{C}^b, \mathbf{C}$)
    **Initialize**
        $\mathbf{B} \in \mathbb{R}^{N \times N} \leftarrow \mathbf{0}$

    // Find the indices of the best and second best counterfactual in $\mathcal{S}^{best}$ for each of the M cost function.
    $\boldsymbol{b}^1 \in \mathbb{R}^M = \arg\max_i \mathbf{C}_{ij}^b$
    $\boldsymbol{b}^2 \in \mathbb{R}^M = \arg \text{ second } \max_i \mathbf{C}_{ij}^b$
    // Iterate over all pairs of counterfactuals.
    **forall** $p, q \in [N] \times [N]$ **do**
        // Iterate over cost functions for which $p^{th}$ counterfactual in $\mathcal{S}^{best}$ has the minimum cost.
        **forall** $r \in \{i \in [M] \mid \boldsymbol{b}_i^1 = p\}$ **do**
            **if** $\mathbf{C}_{pr}^b > \mathbf{C}_{qr}$ **then**
                // This replacement reduces the cost of $\mathcal{S}^{best}$ by $\mathtt{C}_{pr}^b - \mathtt{C}_{qr}$.
                $\mathbf{B}_{pq} += \mathbf{C}_{pr}^b - \mathbf{C}_{qr}$
            **else**
                // $\mathtt{C}_{\boldsymbol{b}_r^2, r}^b$ = cost of second best counterfactual in $\mathcal{S}^{best}$ for $r^{th}$ cost function.
                $\mathbf{B}_{pq} += \mathbf{C}_{pr}^b - min(\mathbf{C}_{qr}, \mathbf{C}_{\boldsymbol{b}_r^2, r}^b)$
        **end**
    **end**
    **return** $B$
**end**

---

Listing 1: Code to Compute Benefits

```python
def computeBenefits(self, best_metrics, curr_metrics):
    best_idx_per_cost = best_metrics.argsort(0)[0]
    second_best_idx_per_cost = best_metrics.argsort(0)[1]
    benefit_matrix = np.zeros((len(best_metrics), len(curr_metrics)))
    for bb, best_met in enumerate(best_metrics):
        for cc, curr_met in enumerate(curr_metrics):
            ben = 0
            for j in np.where(best_idx_per_cost == bb)[0]:
                if best_met[j] > curr_met[j]:
                    ben += (best_met[j] - curr_met[j])
                else:
                    ben += (best_met[j] - min(curr_met[j],
                        best_metrics[second_best_idx_per_cost[j]][j]))
            benefit_matrix[bb, cc] = ben
    return benefit_matrix
```

Listing 2: Code for getting pair to replace

```python
def getReplaceIdx(benefit_matrix) -> List[Tuple]:
    # If there is no benefit on any replacement then just return an empty list.
    if (benefit_matrix > 0).sum() == 0:
        return []
    else:
        # Greedily replace the current cfs with cf from the best-so-far set that leads to the most
        #     benefit. Note that this can be a sub-optimal replacement strategy as it's greedy but
        #     works well in practice.
        org = [] # Indices in the best-so-far cfs set.
        new = [] # Indices of the current set.

        # Iterate over each cfs of the N current set cfs
        for cc in range(len(benefit_matrix)):
            # Iterate over cfs from the best-so-far set in descending order of the benefit from
            #     replacement.
            for idx in benefit_matrix[:, cc].argsort()[::-1]:
                # If best-so-far cf has not already been replaced and has a positive benefit then
                #     add the indices to org and new list.
                if idx not in org and benefit_matrix[idx, cc] > 0:
                    org.append(idx)
                    new.append(cc)
        # create a list of tuples of (org, new) and return.
        return list(zip(org, new))
```

education level from *High-school* to *Masters* then there are two steps involved in the process. First, they need to get a *Bachelors* degree and then a *Masters* degree in which case, the user's cost is proportional to 2 because of the two steps involved in the process.

### B.2.2 Merging Counterfactual Sets

When searching for a good solution set, it would be useful to have the option of improving on the best set we have obtained so far using individual counterfactuals in the next candidate set we see, rather than waiting for a new, higher-scoring set to come along. While optimizing for objectives like diversity, which operate over all pairs of elements in the set, it is computationally complex to evaluate the change in the objective

function if one element of the set is replaced by a new one. To evaluate the change in objective in such cases, we need iterate over all pairs of element in the best and the candidate set and then evaluate the objective for the whole set again. The iteration over both the sets here is not the hard part but the computation that needs to be done within. For our objective, we can compute costs for individual recourses rather than sets, meaning we can do a trivial operation to compute the benefits of each pair replacement. But, if we wanted to do this with diversity then for each pair of replacement we need to compute additional $\mathcal{S}$ distances for each replacement because the distance of the new replace vector needs to be computed with respect to all the other vectors, for each iteration of the nested loop. This quickly makes it infeasible to improve the best set by replacing individual candidates with the best set elements. However, for metrics where it is easy to evaluate the effect of individual elements on the objective function, we can easily merge the best set and any other set $\mathcal{S}_t$ from time $t$ to monotonically increase the objective function value.

In our objective function, EMC, we can compute the goodness of individual counterfactuals with respect to all the Monte Carlo samples (Robert & Casella, 2010). Given a set of counterfactuals we can obtain a matrix of incurred cost $\mathbf{C} \in \mathbb{R}^{N \times M}$, which specifies the cost of each counterfactual for each of the Monte Carlo samples. We can use this to update the best set $\mathcal{S}^{best}$ using elements from the perturbed set $\mathcal{S}_t$ at time $t$. This procedure is defined in algorithm 6. It iterates over all pairs of element in $s_i \in \mathcal{S}^{best}$ and $s_j \in \mathcal{S}_t$ and computes the change that will occur in the objective function by replacing $s_i \to s_j$. Note that we are not recomputing the costs here. Given $\mathcal{S}^{best}$, $\mathcal{S}_t$, $\mathbf{C}^b$ and $\mathbf{C}$, we can guarantee that we will update the best set $\mathcal{S}_{best}$ in a way to improve the mean of the minimum costs incurred for all the Monte Carlo samples. This is shown in algorithm 6 and the monotonicity of the EMC objective under this case can be formally stated as,

**Theorem B.1** (Monotonicity of Cost-Optimized Local Search Algorithm). *Given the best set, $\mathcal{S}_{t-1}^{best} \in \mathbb{R}^{N \times d}$, the candidate counterfactual at iteration $t$, $\mathcal{S}_t \in \mathbb{R}^{N \times d}$, the matrix $\mathbf{C}^b \in \mathbb{R}^{N \times M}$ and $\mathbf{C} \in \mathbb{R}^{N \times M}$ containing the incurred cost of each counterfactual in $\mathcal{S}_{t-1}^{best}$ and $\mathcal{S}_t$ with respect to all the $M$ sampled cost functions $\{\mathcal{C}_i\}_{i=1}^{M}$, there always exist a $\mathcal{S}_t^{best}$ constructed from $\mathcal{S}_{t-1}^{best}$ and $\mathcal{S}_t$ such that*

$$EMC(s_u, \mathcal{S}_t^{best}; \{\mathcal{C}_i\}_{i=1}^{M}) \leq EMC(s_u, \mathcal{S}_{t-1}^{best}; \{\mathcal{C}_i\}_{i=1}^{M})$$

*Proof.* To prove this theorem, we construct a procedure that ensures that the EMC is monotonic. For this procedure, we prove that the monotonicity of EMC holds. Check algorithm 6 for a constructive procedure for this proof, which is more intuitive to understand.

We start off by noting that each element of $\mathbf{C}_{ij}^b$ is the cost of the $i^{th}$ counterfactual $s_i^b$ in the best set $\mathcal{S}_{t-1}^{best}$ with respect to the cost function $\mathcal{C}_j$ given by $\text{Cost}(s_u, s_i^b; \mathcal{C}_j)$. Similarly $\mathbf{C}_{ij} = \text{Cost}(s_u, s_i; \mathcal{C}_j)$ where $s_i$ is the $i^{th}$ candidate counterfactual. Note that, the EMC is the average of the MinCost with respect to all the sampled cost function $\mathcal{C}_j$. What this means is that given a pair of counterfactual from $\mathcal{S}_{t-1}^{best} \times \mathcal{S}_t$ and for each $\mathcal{C}_j$, we can compute the change in the MinCost which we describe later. These replacements can lead to an increase in the cost with respect to certain cost function but the overall reduction depend on the aggregate change over all the cost functions. Given this, for each replacement candidate pair in $\mathcal{S}_{t-1}^{best} \times \mathcal{S}_t$, we can compute the change in EMC by summing up the changes in the MinCost across all cost functions $\mathcal{C}_j$ ; this is called the cost-benefit for this replacement pair. The cost benefit can be negative for certain replacements as well if the candidate counterfactual increases the cost across all the cost functions. The pairs with the highest positive cost benefits are replaced to construct the set $\mathcal{S}_t^{best}$, if no pair has a positive benefit then we keep set $\mathcal{S}_{t-1}^{best} = \mathcal{S}_t^{best}$. Hence, this procedure monotonically reduces EMC. We now specify how the change in MinCost can be computed to complete the proof.

To compute the change in MinCost for a single cost function $\mathcal{C}_i$, first we find the counterfactual in $\mathcal{S}_{t-1}^{best}$ with the lowest and second lowest cost which we denote by $s_{l_1}^b$ and $s_{l_2}^b$. These are the counterfactuals which can affect the MinCost with respect to a particular cost function $\mathcal{C}_i$. This is true because when we replace the counterfactual $s_{l_1}^b$ which has the lowest cost for $\mathcal{C}_j$ with a new candidate counterfactual $s_i$, there are two cases. Either, $\mathbf{C}_{l_1 j}^b > \mathbf{C}_{ij}$ or $\mathbf{C}_{l_1 j}^b \leq \mathbf{C}_{ij}$. In case when the candidate $s_i$ has lower cost for $\mathcal{C}_j$ than $\mathbf{C}_{l_1 j}^b$, i.e. $\mathbf{C}_{l_1 j}^b > \mathbf{C}_{ij}$, then the replacement reduces the cost by $\mathbf{C}_{l_1 j}^b - \mathbf{C}_{ij}$. In case when the candidate cost for $\mathcal{C}_j$, $\mathbf{C}_{ij}$, is higher than the lowest cost in the best set $\mathbf{C}_{l_1 j}^b$, i.e. $\mathbf{C}_{l_1 j}^b \leq \mathbf{C}_{ij}$, it means that this replacement will increase the cost for $\mathcal{C}_i$ by $\mathbf{C}_{l_1 j}^b - min(\mathbf{C}_{ij}, \mathbf{C}_{l_2 j}^b)$. Here, $\mathbf{C}_{l_2 j}^b$ is the second lowest cost counterfactual for $\mathcal{C}_i$. Note that the change in this case will be negative and also depend on the second best counterfactual

because once the $s_{l_1}^b$ is removed from the set, the best cost for $\mathcal{C}_i$ will either be for $s_{l_2}^b$ or $s_i$, hence we take the minimum of those two and then take the difference as the increase in cost. Please refer to Algorithm 6 for a cognitively easier way to understand the proof. □

### B.2.3 Other Methods

In this section, we describe some of the optimization methods used by relevant baselines.

**1. DICE** (Mothilal et al., 2020) perform gradient-based optimization in this continuous space while optimizing for objective defined in Section B.1.1. Their final objective function is defined as

$$C(\boldsymbol{x}) = \operatorname*{arg\,min}_{\boldsymbol{c}_1,\ldots,\boldsymbol{c}_k} \frac{1}{k} \sum_{i=1}^{k} \operatorname{loss}(f(\boldsymbol{c}_i), y) + \frac{\lambda_1}{k} \sum_{i=1}^{k} \operatorname{dist}(\boldsymbol{c}_i, \boldsymbol{x}) - \lambda_2 \operatorname{dpp\_diversity}(\boldsymbol{c}_1, \ldots, \boldsymbol{c}_k)$$

where $\boldsymbol{c}_i$ is a counterfactual, $k$ is the number of counterfactuals, f(.) is the black box ML model, yloss(.) is the metric which minimizes the distance between models prediction and the desired outcome $y$. dpp\_diversity(.) is the diversity metric as defined in Section B.1.1 and $\lambda_1$ and $\lambda_2$ are hyperparameters to balance the components in the objective. Please refer to Mothilal et al. (2020) for more details.

**2. FACE** (Poyiadzi et al., 2020) operates under the idea that to obtain actionable counterfactuals they need to be connected to the user state via paths that are probable under the original data distribution aka high-density path. They construct two different types of graphs based on nearest neighbors (Face-knn) and the $\epsilon$-graph (Face-Eps). They define geodesic distance which trades-off between the path length and the density along this path. Lastly, they use the Shortest Path First Algorithm (Dijkstra's algorithm) to get the final counterfactuals. Please refer to (Poyiadzi et al., 2020) for more details.

**3. Actionable Recourse** (Ustun et al., 2019) tries to find an action set $\boldsymbol{a}$ for a user such that taking the action changes the black-box models decision to the desired outcome class, denoted by +1. They try to minimize the cost incurred by the user while restricting the set of actions within an action set $A(x)$. The set $A(x)$ imposes constraints related to feasibility and actionability with respect to features. They optimize the log-percentile shift objective (see Section B.1.1). Their final optimization equation is

$$\min cost(\boldsymbol{a}; \boldsymbol{x}) \quad s.t. \quad f(\boldsymbol{x} + \boldsymbol{a}) = +1, \ \boldsymbol{a} \in A(\boldsymbol{x})$$

which is cast as an Integer Linear Program (Mittleman, 2018) to provide users with recourses. Their publicly available implementation is limited to a binary case for categorical features,[4] hence we demonstrate results on the binarized version of the dataset.

---

[4]Please refer to the this example where they mention about these restricted abilities https://github.com/ustunb/actionable-recourse/blob/master/examples/ex_01_quickstart.ipynb

