# OpenReview forum: "INSPIRE: Incorporating Diverse Feature Preferences in Recourse"
_TMLR — Accepted by TMLR_

### Review · Reviewer_1jrs · 2023-12-15

**Summary Of Contributions:**

The paper study a new approach for recourse generation. In this problem, there is a model that outputs a decision on the basis of the user's features given in input. The goal is the design of an algorithm that suggests to the user how to change their features in order to change the model decision. The authors propose a novel method for recourse generation that account for diverse features preference. Moreover, they propose a new model that account for the fact that different users have different costs in changing their features. Finally, they propose an optimization method to generate recourses. The proposed method is then evaluated according to a new evaluation procedure.

**Audience:**

Yes

**Claims And Evidence:**

Yes

**Requested Changes:**

I suggest to expand the textual description of Algorithm 1 in Section 4.3.

**Strengths And Weaknesses:**

The paper studies an interesting problem. Moreover, the paper is well written: the problem is clearly described and the contribution are well explained. Finally, the extensive experimental analysis compares the method with previous approach on several aspects, showing the superiority of the proposed method.

The main weakness of the paper is the algorithmic contribution. Indeed, the main contribution of the paper is the proposal of new approaches/models and the algorithmic contribution is limited to Algorithm 1.

---

> ### Author Response · Authors · 2024-01-15
> **Thank you for your time!**
>
> First of all, thank you for appreciating our work and for finding our experiments section to be extensive.
>
> We would like to point out that the contribution of the work is not limited to the COLS method (Algorithm 1) but in proposing the INSPIRE framework to approach algorithmic recourse which has many novel components like the cost function sampling scheme, the new EMC objective, and the new evaluation produces and metrics like FS@k. I hope this helps to position our work better and to highlight its novelty.
>
> As per your suggestion, we have updated Section 4.3 of the main paper to contain more details about Algorithm 1.
>
> **I hope this addressed all of your questions and requested changes. Please do let us know in case there are any other clarifications needed.**

---

### Review · Reviewer_hHv7 · 2024-01-11

**Summary Of Contributions:**

The focus of the paper is on the longstanding problem in fair ML of "actionable recourse". If a classification system is to be used to allow or deny access of users to scarce resources (paradigmatically a credit decision) based on some features, then it may be desirable or even legally necessary to tell them how they could change their features in order to change the decision from negative to positive. It may also be desirable to ensure that the cost to the user is sufficiently small (feasible), according to some cost model where changing certain features may be more or less difficult.

The authors first point out that there is no reason to assume that every user has the same cost function; moreover, users' cost functions are likely unknown (even to themselves). So the authors present a technique that they call INSPIRE which involves sampling a large number of possible cost functions, then generating a recourse set in order to minimize the mean (over each cost function) of the cost of the best recourse for that function. This is a difficult optimization problem for many reasons, and they propose a local search procedure to tackle it. The sampling scheme, the new objective, and the local search procedure together constitute INSPIRE.

The authors then present a set of experiments on the classic Adult and COMPAS datasets. They assume that each simulated user in this dataset has some unknown ground truth cost function, which will be used to measure performance but not made available to any of the recourse methods. They measure the quality of generated recourses in terms of a number of cost metrics based on this ground truth as well as other standard distance metrics. They consider a number of "ablation studies" to show that their new contributions really do improve performance, and they consider robustness to various incorrect modeling choices as well as measuring impacts on demographic fairness.

**Audience:**

Yes

**Broader Impact Concerns:**

The authors include a broader impact statement in appendix A.1 which seems to sufficiently address concerns with this general type of work. I tentatively agree with the authors' claim that work on actionable recourse is a "robust good". I don't think the novel contributions of the work introduce any novel issues of broader impact.

**Claims And Evidence:**

Yes

**Requested Changes:**

I think more discussion of modeling choices about generating the sampled cost functions, and how a "real-world" practitioner might go about this (perhaps in scenarios with different levels of information about the true user preferences) would be helpful, though not essential.

I also weakly speculate that tools from robust and distributionally robust optimization might be useful for this problem, and some discussion about these optimization frameworks may be worthwhile.

**Strengths And Weaknesses:**

Strengths:
- The issues with actionable recourse that the paper addresses are compelling and worthy of study.
- Given sampled cost functions, the idea of expected minimum cost seems like the "right" way to do things under their model.
- The result of this model is a really gnarly optimization problem to find the recourse options and their COLS procedure seems like a useful way to attack it.
- The experiments seemed to me to be convincing, including ablation studies showing that while both of their core contributions give some performance improvement, it is the use of the expected minimum cost objective that makes the bigger contribution.


Weaknesses:
- The method for generating cost functions seems fine, but also somewhat arbitrary. It's not clear that this "really" captures what user preferences would look like in any meaningful sense. On the other hand, that type of "realism" may be outside the scope of the paper, which only works on standard datasets anyway, and more of a modeling question for someone who wants to use this method in the real world.
Further moderating the above weakness somewhat, the authors do show some experiments where even when the train-time distribution is misspecified, their approach still is able to outperform baselines.

---

> ### Author Response · Authors · 2024-01-15
> **Thank you for your time!**
>
> First of all, thank you for understanding the contributions of our work as a framework as opposed to just being a method and for understanding the complexity of estimating meaningful cost functions in the absence of any user preferences.
>
>
>
> **[Requested Change 1 and Weakness 1] I think more discussion of modeling choices about generating the sampled cost functions, and how a "real-world" practitioner might go about this ....**
>
> Given this complexity, we designed our cost function sampling procedure in a way that it can incorporate partial feedback from the real user whenever available. This means that to obtain the best estimate of a user's cost function, a real-world practitioner can design a system where they can ask each user to provide either the features (F_p) that they find easy to change (which can be converted to preference scores using Algorithm 4) or directly provide a preference score (p) for each feature if they can. Given either of these our Algorithm 5 can be used to sample cost functions that are a better estimate of the user's real cost function. Note that these cost functions are still estimates and try to capture different ways a user can think of cost, i.e. in terms of percentiles or Euclidian (absolute change in feature). The practitioner can use these more aligned samples to optimize the EMC objective to generate better recourse for the user.
>
> Throughout the paper, we tackle the hardest cases and assume that such user preference information is not directly available as collecting such preferences is hard as noted by Rawal & Lakkaraju (2020). We wanted our cost function to account for the two popularly used ideas of cost, percentile, and Euclidean cost and hence we use a mixing coefficient (\alpha) that allows us to modulate the effect of each of these in our final cost function. Moreover, the cost function samples are generated by randomly sampling multiple random feature preferences, and hence by design, it accounts for all the possible preferences different users can have. This allows us to cater to the users for whom we don't have any preference information.
>
> I hope this clarifies your doubts and we have added a version of this in Section 5.1 in the main paper.
>
>
> **[Requested Change 2] Tools from robust and distributionally robust optimization might be useful for this problem ...**
>
> Thank you for pointing this out! Indeed it seems like ideas from distributionally robust optimization might be relevant to our problem. In the current framing of the EMC objective function, it is not directly clear to us how DRO can be used. However, there might be alternative objective functions that could adhere to the DRO framework and still be useful for the recourse generation problem under consideration. We have added a very short discussion about it in the related work section and left it to future works to fill in the gaps.
>
> **I hope this addressed all of your questions and requested changes. Please do let us know in case there are any other clarifications needed.**

---

### Review · Reviewer_btjX · 2024-01-25

**Summary Of Contributions:**

This paper deals with the generation of counterfactual examples (CF) that take into account the cost of change to the user. The proposed approach addresses the situation where the cost function varies from user to user and is unknown. The solution is not to propose a single CF, but a set of CFs, so that the user can find a suitable CF from this set.

The paper introduces a new criterion to be optimized for selecting a set of relevant CFs, as well as an algorithm (COLS) for optimizing this criterion. The paper compares the quality of the CFs provided by this approach with that of CFs provided by other state-of-the-art approaches (DICE, FACE, and Actionable Recourse).

**Audience:**

Yes

**Broader Impact Concerns:**

Nothing to mention

**Claims And Evidence:**

Yes

**Requested Changes:**

(critical) Clarify the behavior of `getReplaceIdx()` in Algorithm 1.

(greatly recommended) Clarify, in the main paper, the definition of Algorithms 2 and 3

(greatly recommended) Clarify the high-level presentation of Algorithm 3. Consider chaging its name.

Consider changing the metrics in the experiments.

Typos
- legend of Figure 1: "... does well [under a for] certain types ..."
- bottom of page 2: a space is missing after the sentence ending with "(See Figure 1)."
- top of page 3: " ... this setup can be used to compare and [rank order] different methods..."
- Section 4.1: one occurence of "FP" has been misspelled by "PF"

**Strengths And Weaknesses:**

## Strengths
The paper addresses a problem of practical interest.

The proposed optimization criterion is simple and intuitive: search for a set of CFs such that any cost function (in a large set) is small for at least one of the CFs. Despite this simplicity, in experiments this criterion leads to better sets than DICE, which optimizes a diversity criterion.

## Weaknesses
The metrics introduced for the experiments can be misleading.
> One of the criteria for a CF example is its closeness with respect to the original example, either to make it actionable, or to make it understandable to the user why the original example is not in the class of the CF. In the current paper, this criterion is evaluated by *proximity*, *population average cost* (PAC), *user coverage* (Cov), and *fraction of users satisfied* (FS@k). All these metrics are defined based on an average criterion over the set of proposed CFs, which is a misleading criterion. Typically, if the algorithm proposes a set that contains a unique valid CF (an example that is classified differently from the original), even if this CF is far from the original example, these metrics can be made good by adding to this set a large number of non-valid examples (examples that are classified as the original one) that are extremely close to the original. It would be more appropriate to define these metrics as averaged only over valid CFs.
>
> This risk is mitigated in the paper by limiting the size of the sets to 10, but 9 non-valid examples are already sufficient to drastically reduce *proximity*, PAC, Cov, and FS@k.


The presentation of some algorithms could be clarified.

> The feature cost function for ordered variables drastically differs from the one for unordered variables. This point is actually not mentioned in the main paper, the reader as to look at Alg 2 and 3 (in Appendix) to be aware of it. The main paper should at least contain a warning.
>
> Also, the term "Euclidean" is overused for Algorithm 3. This cost function does not depend on the Euclidean distance between two points, but on the number of points in some intervals. As in Algorithm 2, this cost function refers to quantiles, except that Algorithm 3 is limited to the distribution of points either above or below the value of the original example.
>
> Finally, Theorem 4.1 is only valid if there is one unique replacement per iteration in Algorithm 1. However, the current pseudocode of Algorithm 1 is unclear on this point.

---

> ### Author Response · Authors · 2024-01-25
> **Thank you for your time!**
>
> First of all, thank you for understanding the contributions of our work and acknowledging the practical utility of the problem.
>
>
> **Weakness 1 and Requested Changes 3: The metrics introduced for the experiments can be misleading.**
>
> Thanks for the question, we would like to clarify that all the metrics are computed only over the valid CFs. Once we perturb the set we compute the class of each cf and assign a validity flag (i.e. if it is the desired class) to each cf. Then for all future computations (including metrics, replacement, etc) we only work with the valid cfs. Hence, when computing any cost-related metrics, the invalid samples are pre-filtered. We have clarified this in the paper.
>
> Just for reviewer satisfaction, this can be observed in our code as well which is attached as supplementary material. For distance-based metrics please refer to src.utils.metrics.MetricsEvaluator.get_single_sample_metrics, Lines 55-59. For cost-based metrics please refer to src.utils.metrics.UserEvaluator.get_metrics, Lines 229-234. Moreover, the invalid cfs are not used during any computations related to computing benefits, or replacements of best-set elements, this can be seen at src.explainers.search_method.BaseSearch class.cost_local_search, L262-270.
>
> Lastly, from Table 1, we can observe that the validity of the cfs set from COLS/P-COLS is above 90% as well.
>
> We have updated the paper to clarify this confusion.
>
>
> **Weakness2 (point1): The feature cost function for ordered variables drastically differs from the one for unordered variables.....The main paper should at least contain a warning..**
>
> We have clarified this in section 4.1 and have added all the factors that affect the transition cost and have again pointed the reader to the Algorithms.
>
> **Weakness2 (point2) and Requested Changes 3: Also, the term "Euclidean" is overused for Algorithm 3. ... Consider changing its name ...**
>
> As requested by the reviewer, we have changed the name of Euclidean-based cost to Step-based cost throughout the paper with other related changes. We have further clarified the presentation of Algorithm 3 in section 4.1 as well.
>
>
> **Weakness2 (point3): Finally, Theorem 4.1 is only valid if there is one unique replacement ....**
>
> We would like to clarify that Theorem 4.1 states that the EMC(S^best_{t}) <= EMC(S^best_{t-1}), which implies that the cost of the S^best_t is lower than S^best_{t-1}. Our get_replace_idx function only performs swapping of a cf from S^best_{t-1} only if the cf that is being added leads to a cost-benefit, i.e. reduces the overall cost. Moreover, this is true for each and every swap that happens otherwise the swap is not done. Hence, after performing the swaps to obtain S^best_t the EMC(S^best_{t}) < = EMC(S^best_{t-1}) by design.
>
> The additional details of getReplaceIdx might help to clarify this further. Please let us know if there are any further confusions. We also looking forward to resolving all of your queries.
>
>
> **Requested Changes 1: Clarify the behavior of getReplaceIdx() in Algorithm 1.**
>
> We have updated Algorithm 1 to add the details of the getReplaceIdx function and discussed it in section 4.3 as well. Moreover, we have added Code Block 1 and Code Block 2 which provide the code for computeBenefits and getReplaceIdx functions to better understand their internals.
>
>
> **Requested Changes 2: Clarify, in the main paper, the definition of Algorithms 2 and 3**
>
> As requested, we have updated section 4.1, to further clarify the definition of Algorithms 2 and 3. For easier understanding, we have also added an example to describe both types of transition costs (percentile and Euclidean).
>
>
>
> **We are looking forward to an engaging discussion. If you have any other questions or requested changes then please do let us know!**

---

> ### Comment · Reviewer_btjX · 2024-02-22
>
> Thank you for these precise answers and the corresponding additions and clarifications to the article. I'm reassured on most of the points I raised.
>
> However, I disagree with the part of the code that improves $\cal{S}^{best}$. The improvement matrix $B$ should be recomputed after each modification of a point in $\cal{S}^{best}$, since this modification can change the arg_max and arg_second_max for several cost functions.

---

> > ### Author Response · Authors · 2024-02-28
> > **Thank you for your reply!**
> >
> > We are glad that we can address the concerns raised by you. Thank you so much for taking the time to respond to our rebuttal.
> >
> > > The improvement matrix should be recomputed after each modification of a point in, since this modification can change the arg_max and arg_second_max for several cost functions.
> >
> > **Answer:** As mentioned in the paper,  "We then greedily replace the CFs in $\mathcal{S}\^{best}\_{t−1}$ with cfs from $\mathcal{S}\_{t}$ to obtain $\mathcal{S}\^{best}\_{t}$". Hence the procedure to update the best set is not optimal as for many cf in $\mathcal{S}\^{best}\_{t−1}$ as the top or the second top replacement in $\mathcal{S}\_{t}$ might have already been used to replace with some other cf before it in $\mathcal{S}\^{best}\_{t−1}$ and for such cf in the $\mathcal{S}\^{best}\_{t−1}$ no replacement is done (shown in alg 1; line with if $bestIdx \notin replaced$).
> >
> > However, in the current procedure, a replacement is only done if there is a benefit and hence the current procedure still guarantees that the updated set $\mathcal{S}\^{best}\_{t}$ is at least as good as the last best cfs set $\mathcal{S}\^{best}\_{t-1}$.
> >
> > We are looking forward to any other questions you might have.

---

### Author Response · Authors · 2024-01-25
**General Response to all Reviewers | Summary of Changes in the PDF**

Dear Reviewers, thanks for putting in your time to provide us with valuable comments and suggestions to make our paper better. Following your recommendation, we have updated the paper to address all of your concerns.

List of major changes.

1. **Changed the name of Euclidean Cost to Steps-based Cost** and related changes.
2. **Sec 4.1 More details on Algorithms 2 and 3 and how they model feature properties.** Expanded on the different properties a feature can have (ordered/unordered), (mutable/conditionally-mutable/immutable), etc, and added a discussion on how cost function model them using Algorithms 2 and 3.
3. **Sec 4.1 Additional details on the two types of transition cost** (percentile and steps/euclidean) and provided examples for ease of understanding.
4. **Sec 4.3: Additional details of COLS and Algorithm1** Clarifies question related to the validity of cfs, expanded getReplaceIdx function in Algorithm1, and added details of ComputeBenefits and getReplaceIdx in the text along with code block in the appendix.
5. **Sec 5.2: Added Discussion on how real-world practitioners can define and sample cost functions** in the presence or absence of preference data from the user.

Please let us know if any other changes can improve the quality of our paper.

---

### Decision · Action_Editor_mrV4 · 2024-03-07

**Recommendation:** Accept with minor revision

**Comment:**

All reviewers recommended acceptance of the submitted paper or leaned towards acceptance. The authors also addressed all the issues and concerns raised by the reviewers. Given that the topic is relevant and the paper makes counterfactual generation more realistic/useful, I follow their recommendations. Nevertheless, upon a final reading of the paper, I noticed several glitches in the paper that the authors should finally address (sorry for partly being picky):
* Improve the description of Algorithm 1. Present the algorithm as general as possible, i.e., do not fix hamDist to 2 (or is there a reason for doing so?). There are references to "Code 1" and "Code Block 2" which are unclear if one prints the paper (what they refer to only becomes clear through the links in the pdf); the update to $S^{best}$ is hard to parse given the used notation - can this be put abstractly and more understandably; it is not specified how you pick the hamDist=2 features you change - I assume you pick them uniformly at random? In the return line, $C^b$ is set differently than before.
* Verify that Theorem 4.1 is still consistent with the updated Algorithm 1.
* "we propose a genetic algorithm Mitchell (1998)" => put citation in brackets or similar
* $F = {f1, f2, ...fh}$ => $F = {f1, f2, ..., fh}$
* "there is no particular ordering of the feature values in terms of difficulty" => "difficulty" is unclear.
* "where s^f is the value of feature f in the state vector" => s^f should not be bold; end sentence with $\mathbf{s}$
* In Eq. (1) you spell out the constraints while you omit them in Eq. (4) => unify
* "where F is the black-box ML model and 1 is the desired class" => clarify, up to this point the ML model wasn't mentioned much, that it is black-box was mentioned at all (also the relevance is unclear); clarify what a desired class is and that there are only 2 classes (?)
* "hence we use the EMC objective account for user FP" => add a forward reference
* There are several more typos (capitalization, commas) in the newly added text, e.g., in section 4.1. Please revise those carefully.

**Audience:**

Yes, the topic is of relevance to a subcommunity of TMLR's audience.

**Claims And Evidence:**

The submitted paper makes several claims regarding more realistic and fair recourse generation which are well supported by arguments and also experiments.

---

> ### Author Response · Authors · 2024-03-17
> **Thanks for your hard work!**
>
> First of all, we would like to thank you for your hard work.
>
> We have uploaded the camera-ready version of the paper and addressed most of your comments to the best of our understanding. However, there was one comment that was not clear to us and we have omitted edits on it, if you can clarify what you mean by your fourth comment "F= f1,f2, ..., fh -> F= f1,f2, ..., fh" we will update that further as well.
>
>
> Thanks again!